# Genome graphs reveal the importance of structural variation in *Mycobacterium tuberculosis* evolution and drug resistance

Aleix Canalda-Baltrons [1], Matthew Silcocks [1], Michael B. Hall [2], Derrick Theys[1], Xuling Chang 常戌灵 [1,3,4], Linda T. Viberg [5], Norelle L. Sherry [6,7], Lachlan Coin [2] & Sarah J. Dunstan [1] ✉

Structural variants (SVs) are increasingly recognized as key drivers of bacterial evolution, yet their role has not been explored thoroughly. This is due to limitations in traditional short-read sequencing and linear reference-based analyses, which can miss complex structural changes. Tuberculosis (TB), a disease caused by *Mycobacterium tuberculosis* (*Mtb*), remains a major global health concern. In this study, we harness long-read sequencing technologies and genome graph tools to construct a *Mtb* pangenome reference graph (PRG) from 859 high-quality, diverse, long-read assemblies. To enable accurate genotyping of SVs leveraging the PRG, we developed miniwalk, a tool that outperforms a traditional linear genome-based approach in precision for SV detection. We characterize patterns of structural variation genome-wide, revealing a virulence-associated ESX-5 deletion to be recurrent across the phylogeny, and fixed in a sub-lineage of L4. Systematic screens for additional genes that are recurrently affected by SVs implicated those related to metal homeostasis, including a copper exporter fixed in the widely distributed L1.2.1 sub-lineage. Lastly, we genotyped 41,134 isolates and found SVs putatively associated with resistance to various first and second-line drugs. These findings underscore the broader role of SVs in shaping *Mtb* diversity, highlighting their importance in both understanding evolution and designing strategies to combat drug-resistant TB.

*Mycobacterium tuberculosis* (*Mtb*) was the cause of 10.8 million new cases of tuberculosis (TB) and 1.25 million deaths in 2023 alone[1]. Despite the availability of effective antibiotics, drug-resistant TB is widespread, with 400,000 new drug-resistant cases recorded in 2023, leading to poorer patient outcomes[1]. Next-generation sequencing (NGS) technologies have proven useful in predicting drug resistance from genomic sequences of *Mtb* and are being implemented in public health laboratories in some jurisdictions[2,3].

Genetic variability within the *Mtb* genome can reveal aspects of the pathogen's biology, and single-nucleotide polymorphisms (SNPs)

[1]Department of Infectious Diseases, The University of Melbourne at the Peter Doherty Institute for Infection and Immunity, Melbourne, VIC, Australia. [2]Department of Microbiology and Immunology, The University of Melbourne at the Peter Doherty Institute for Infection and Immunity, Melbourne, VIC, Australia. [3]Department of Paediatrics, Yong Loo Lin School of Medicine, National University of Singapore, Singapore, Singapore. [4]Khoo Teck Puat - National University Children's Medical Institute, National University of Singapore, Singapore, Singapore. [5]Victorian Infectious Diseases Reference Laboratory, Melbourne Health at the Peter Doherty Institute for Infection and Immunity, Melbourne, VIC, Australia. [6]Microbiological Diagnostic Unit Public Health Laboratory (MDU PHL), The University of Melbourne at the Peter Doherty Institute for Infection and Immunity, Melbourne, VIC, Australia. [7]Department of Infectious Diseases and Immunology, Austin Health, Melbourne, VIC, Australia. ✉e-mail: sarah.dunstan@unimelb.edu.au

have been investigated for this purpose. SNPs in *Mtb* have been well characterized, are used to define lineages[4] and constitute the majority of mutations listed by the World Health Organization (WHO) responsible for drug resistance (DR), alongside small insertions and deletions (indels; genomic variants smaller than 50 bp)[5]. Structural variants (SVs), that is, genomic variants larger than 49 bp, have been less well studied.

SVs have played an important role in various aspects of TB pathogenesis. The TbD1 deletion has been reported to be involved in the evolution and differentiation of the modern from the ancestral *Mtb* lineages, with modern lineages gaining enhanced resistance to hypoxic conditions[6]. Various deletions have also been associated with resistance to anti-TB drugs, for example, the deletion of *katG* to isoniazid resistance or the deletion of *ethA* to ethionamide resistance[7].

Despite their potential significance, SVs in *Mtb* have been less comprehensively studied than SNPs and are often dismissed in studies[8,9]. This knowledge gap is partly due to the limitations of short-read sequencing technologies (Illumina), which can miss or mischaracterize SVs—particularly in repetitive or GC-rich regions—as well as technical and bioinformatics limitations in variant detection and interpretation[10]. Long-read sequencing technologies (Oxford Nanopore Technologies (ONT) or Pacific Biosciences (PB)) have provided an opportunity for accurate identification of SVs, enabling the exploration of their role in DR and *Mtb* pathogenesis[11–14]. However, Illumina remains the most used technology, with a total of ~57,000 new NCBI entries in 2024 against ~2250 for ONT and PB.

Pangenome reference graphs (PRGs) are data structures that represent both shared and variable genomic sequences across multiple individuals or isolates[15]. Unlike traditional linear reference genomes, which provide a single consensus sequence, PRGs can capture a broader spectrum of genomic diversity. This allows for improved detection of structural variants (SVs), especially when analyzing short-read sequencing data[16]. Although genome graph tools have been scarcely exploited in *Mtb* genomics[14,17], they hold the potential to unveil a wider scope of SVs across thousands of isolates that only have Illumina data available.

In this study, we leverage long-read sequencing technologies and genome graph tools to create an *Mtb*-PRG. We also develop miniwalk, a tool that genotypes SVs from an assembly mapped to a PRG and shows it is significantly more precise than genotyping using a linear genome. We also explore the evolution of SVs across the phylogeny, identifying recurrent mutations implicating the ESX loci, and a key copper exporter now deleted in a sub-lineage of L1. Lastly, we genotype SVs in 41,134 isolates to search for associations with DR across 22 drugs. Our findings underscore the diverse roles of SVs in shaping *Mtb* evolution and contributing to DR-TB.

## Results

### Construction of a *M. tuberculosis* pangenome reference graph
We obtained 1363 long-read datasets (ONT or PB) from public repositories and performed quality control for read length and decontamination of human DNA before assembling genomes with Flye[18]. After also incorporating 176 long-read assemblies from NCBI to enlarge the dataset, all assemblies were evaluated with QUAST[19], with 834 remaining after filtering (see Methods; Supplementary Fig. 1). To ensure phylogenetic coverage, we sequenced 25 isolates from underrepresented lineages with ONT, achieving representation of lineages 1–9, including phenotypically drug-resistant isolates without known resistance mutations (Supplementary Table 1). Sub-lineage 2.2.1 (Beijing) was the most frequent of the dataset (23.4%), while lineage 4 and its sub-lineages dominated overall (48%; Supplementary Fig. 2). The final cohort of 859 assemblies, representing both genetic and geographical diversity (Supplementary Data 1 and Supplementary Fig. 3), had a mean N50 of 2.13 Mb (range: 11.2 kb–4.45 Mb) and mean

coverage depth of 125× (range: 7 × -859 × ; full metrics in Supplementary Data 2).

To construct the *Mtb*-PRG, we used minigraph[20], which efficiently incorporates sequences that are larger than 49 bp and missing from the reference genome while preserving reference coordinates. Starting with the H37Rv reference genome (accession: NC_000962.3) as the backbone, additional genomes were added to the *Mtb*-PRG in lexicographic order by sample name. Newly added sequences formed nodes, which represent distinct segments of genomic sequences that are connected by paths, resulting in 9350 nodes in total. Nodes connected to each other but not part of the reference formed 848 bubbles, with the largest spanning 97,162 bp (H37Rv coordinates: 3,386,328–3,483,490; Supplementary Fig. 4).

### Genotyping SVs using the *Mtb*-PRG and miniwalk is more precise than using a traditional short-read SV caller
PRG-involving approaches have been shown to call SVs more accurately from short-read sequencing data compared to traditional single-reference genome approaches[16]. Since most *Mtb* genomic data are derived from Illumina short-read sequencing, accurately identifying SVs using such data is valuable. However, no existing method genotypes SVs directly from graphs generated using minigraph. To address this, we developed miniwalk, a tool that identifies SVs by comparing the graph traversal paths of an assembly or long reads, to a reference genome mapped to a minigraph PRG (Supplementary Methods 1). Miniwalk outputs insertions (INSs), deletions (DELs), duplications (DUPs), and inversions (INVs).

We benchmarked miniwalk using 16 samples, each of which had a high-quality, polished long-read assembly to act as truth. To avoid bias, the 16 samples were removed from the PRG for the benchmarking[21]. Truth SVs were called using SVIM-ASM[22] on assemblies mapped to the H37Rv reference genome, while miniwalk called/queried SVs on assemblies mapped to the *Mtb*-PRG. Precision and recall, calculated using the miniwalk bench module (Supplementary Methods 2), averaged 0.85 and 0.82, respectively, showing comparable genotyping performance to SVIM-ASM (Fig. 1a). Nevertheless, precision varied across SV types, with DELs of size 500–1000 bp having the lowest precision (0.5; Fig. 1b), though fewer SVs were found in that size range. Further analysis revealed that this drop was largely driven by a single complex SV present in 11 assemblies, consistently detected by miniwalk but missed by SVIM-ASM (Supplementary Fig. 5). We also constructed an additional PRG excluding assemblies with N50 <90,000 bp and coverage <25 × to determine whether a more stringent approach should have been taken. The average precision and recall using the more stringent PRG was 0.84 and 0.83, respectively, indicating that our initial PRG had better precision. To test whether assembly ordering introduced bias, we rebuilt the *Mtb*-PRG with assemblies ordered by lineage rather than lexicographic filename and re-evaluated SV genotyping performance. Precision and recall did not differ significantly (two-sided *t*-tests; precision $p = 0.96$, recall $p = 0.29$), indicating that graph construction order had no measurable effect on genotyping accuracy (Supplementary Fig. 6). To complement real-data benchmarking, we simulated an assembly by spiking real SVs into the H37Rv reference. Miniwalk achieved 0.92 precision and 0.82 recall, outperforming results on real data (0.78/0.81), highlighting the reduced complexity of simulations.

To assess short-read data performance, Illumina reads used for polishing the 16 assemblies were analyzed using manta[23], a leading short-read SV caller[24]. Reads were also assembled with shovill[25], mapped to the *Mtb*-PRG, and SVs genotyped with miniwalk. To assess precision and recall of SV genotyping using these two approaches (linear H37Rv/manta and *Mtb*-PRG/miniwalk), we compared both to SVIM-ASM truth SVs as in Fig. 1a. We found miniwalk demonstrated significantly higher precision (0.7 vs. 0.46 for manta; two-sided *t*-test, $p = 1e-6$) at the cost of slightly lower recall (0.34 vs. 0.42 for manta;

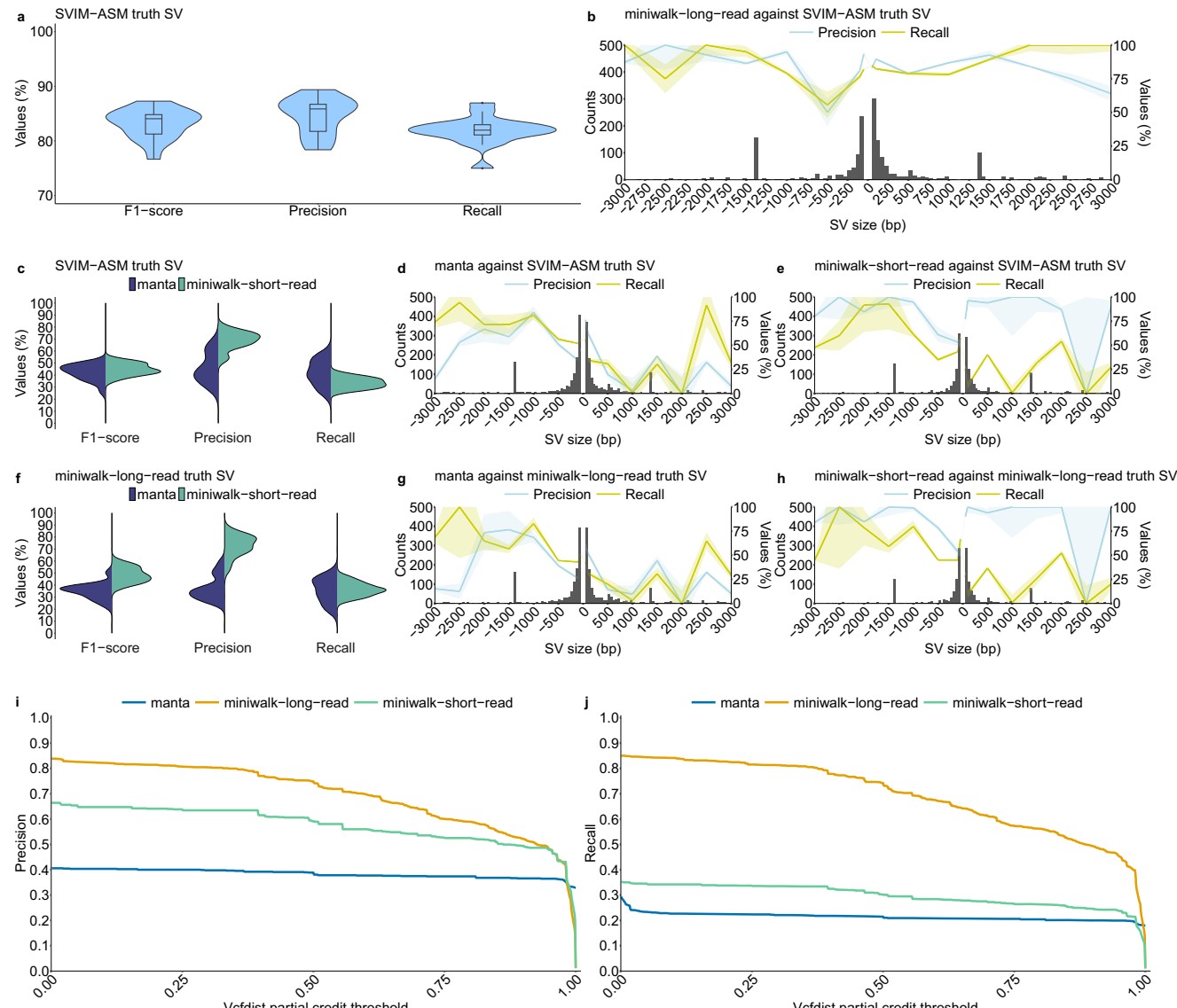

**Fig. 1 | Benchmark of SV genotyping using the *Mtb*-PRG with miniwalk against the linear genome with SVIM-ASM and manta. a** Results comparing *Mtb*-PRG and miniwalk SV genotyping against SVIM-ASM truth SV calls using the 16 independent long-read, polished assemblies. Each data point represents an SV call set from one assembly, with assemblies serving as independent biological replicates. No technical replicates were used. No explicit control group was included; instead, distributions reflect genotyping performance across the independent assemblies. Box plots indicate the median (center line), interquartile range (box), and whiskers extending to 1.5 × the interquartile range; outliers are plotted individually. **b** Histogram with 50 bp bins showing the found SVs in (**a**), with two segments with 50 bp bin from SVs of size 50–100 bp, 400 bp bin for SVs of size 100–500 bp, 500 bp bin for SVs of size 500–3000 bp and SVs larger than 3000 bp grouped together. The two segments show the precision and recall means across different SV sizes. Shaded regions represent 95% confidence intervals (mean ± 1.96 × standard error). **c** Results for genotyping SVs from Illumina data mapped to H37Rv and calling with

manta and mapping to the *Mtb*-PRG and calling with miniwalk against the SVIM-ASM truth SVs in (**a**). **d** Average SV precision, recall and frequency of short-read data stratified by length from manta SVs against the SVIM-ASM truth SVs. Bin sizes as in (**b**). Shaded regions represent 95% confidence intervals (mean ± 1.96 × standard error). **e** Same plot as (**d**) for genotyped short-read SVs using the *Mtb*-PRG and miniwalk against the SVIM-ASM truth SVs. **f** Results for genotyping SVs from Illumina data mapped to H37Rv and calling with manta and mapping to the *Mtb*-PRG and calling with miniwalk-long-read SVs as truth (**a**). **g** Same plot as (**d**) with miniwalk-long-read SVs as truth. **h** Same plot as (**e**) with miniwalk-long-read as truth SVs. **i** Precision values across different vcfdist partial credit thresholds for miniwalk-long-read (long-read assemblies mapped to the graph, genotyped with miniwalk), manta (short reads mapped to H37Rv, genotyped with manta) and miniwalk-short-read (short-read assemblies mapped to the graph, genotyped with miniwalk) SVs against the SVIM-ASM truth SVs. **j** Similar to (**i**), for recall values.

two-sided *t*-test, *p* = 0.007; Fig. 1c). Breaking down by SV type, manta achieved a precision and recall of 0.42 and 0.63, respectively, for DELs and 0.42 and 0.36, respectively, for INSs (Fig. 1d). Meanwhile, miniwalk achieved a precision and recall of 0.64 and 0.54, respectively, for DELs, and 0.86 and 0.19, respectively, for INSs. DELs sized >1000 bp had precision exceeding 0.8 (Fig. 1e). Considering SVs genotyped with miniwalk from the long-read assemblies mapped to the *Mtb*-PRG as the truth (SVs from Fig. 1a, b; miniwalk-long-read truth SV), precision and

recall decreased for Illumina data mapped to the H37Rv linear reference (Fig. 1f, g) but improved slightly for short-read assemblies mapped to the *Mtb*-PRG (Fig. 1f, h). These results highlight miniwalk's precision advantage when using a PRG compared to a linear genome-based method. To explore whether recall could be improved, we created another *Mtb*-PRG, retaining SVs ≥20bp, and evaluated it against the short-read assemblies. We observed, on average, a 2% increase in recall at the cost of a 2% drop in precision (Supplementary Fig. 7).

To further validate the use of miniwalk, we used partial credit from vcfdist[26] to evaluate the similarity between our variant calls and the SVIM-ASM truth set. Vcfdist assigns each pair of variants a partial credit score ranging from 0 to 1, based on edit distance (sequence content and position), where 1 means identical variants and 0 means no overlap. We applied growing partial credit (ct) thresholds starting from 0.001. Lower ct thresholds allow called SVs with partial similarity to the truth to be considered true positives, while higher ct thresholds impose stricter criteria, requiring close matches in SV type, position, and sequence content. This evaluation included the previous long- and short-read assemblies mapped to the *Mtb*-PRG as well as the variants called from manta. For miniwalk-long-read SVs, precision and recall showed a gradual decline as the threshold increased, reflecting differences in sequence content (e.g., long-read sequencing errors, SNPs) and positions between called and truth SVs. Nonetheless, precision (0.75) and recall (0.74) remained moderately high for long-read assemblies mapped to the *Mtb*-PRG at a ct threshold of 0.5, demonstrating the accuracy of SVs called using our graph-based approach. For short-read data, miniwalk outperformed manta across all ct thresholds up to 0.985 for both precision and recall (Fig. 1i, j).

Further analysis revealed that SVs in complex regions were the ones driving the decrease in precision and recall in short-read data (Supplementary Note 1).

These results suggest using a genome graph as a superior alternative to a linear reference genome for genotyping SVs. With this in mind, we proceeded to learn about SVs in *Mtb* using the *Mtb*-PRG and miniwalk.

## Landscape of structural variation across the *M. tuberculosis* genome and phylogeny

We sought to improve the characterization of SVs across diverse *Mtb* lineages using high-quality long-read data. Assemblies used to construct the *Mtb*-PRG were mapped back to the PRG. The 38 most fragmented assemblies were excluded from this analysis as they could not be mapped using minigraph's default parameters (see Methods). SVs in the remaining assemblies (821) were genotyped against the H37Rv reference genome using miniwalk. To further define the diversity of SVs in the *Mtb* genome, DELs and DUPs in close proximity were grouped as copy number variants (CNVs). SVs within ±25bp proximity, ±50 bp size, and of the same type were considered identical across isolates.

A total of 3077 unique SVs were identified relative to H37Rv, broken down into 1127 INSs, 1694 DELs, 59 DUPs, 58 INVs, and 139 CNVs. The largest SV was a 66,582bp region (H37Rv coordinates: 2,901,855–2,968,437) that was inverted and translocated to a novel genomic position (H37Rv coordinates: 415,370) in lineage 9 (Supplementary Fig. 8). The translocation implicated 70 genes and truncated *Rv2577* and *Rv0345*. This rearrangement highlights the complexity of SV patterns observed in *Mtb*, though most isolates presented small DELs, CNVs and numerous IS6110 elements (~1358bp; Fig. 2a). We found various INS-DEL pairs of similar size (≥90%) and high identity (≥90%) across each genome (range of 0–3 per genome), likely stemming from IS6110 transposition. A principal component analysis (PCA) performed with the SV genotypes revealed that SVs contribute to population structure, especially along PC2 which separates the ancient (L1, L5–9) from the modern (L2–4) lineages (Fig. 2b). Notably, the "TbD1" SV at position 1,761,789 explains 2% of the variability in PC2 (Supplementary Fig. 9). The allele frequency spectrum of SVs revealed that over half (1617/3077) were singletons (Supplementary Fig. 10a). Importantly, singleton SVs were not enriched in fragmented assemblies, suggesting they represent genuine rare variation rather than artifacts (Supplementary Fig. 10b).

To understand the acquisition of SVs across lineages over time, SVs were genotyped against an ancestrally reconstructed sequence (MTBC0[27]) and mapped to a phylogenetic tree from the genotyped isolates (Fig. 2c). This showed that all isolates have acquired a similar amount of SVs across the phylogeny, the least being L4 with an average of 67 SVs per isolate and the most being L8 with 82 SVs (though 1 isolate was used). Interestingly, Beijing lineage isolates segregated into two branches, one with an average of 86 SVs per isolate and the other with an average of 68 SVs per isolate, indicating distinct evolutionary trajectories of SV accumulation within this sub-lineage. When analyzing L4 sub-lineages, the globally spread generalists L4.1.2 and L4.3[28] had an average of 60 and 63 SVs per isolate, respectively, whereas the specialists L4.1.3 and L4.6.2[28] had an average of 74. From these results, generalist lineages have significantly less SVs than specialist lineages (two-sided Wilcoxon test, *p* value = 0.03). SVs are generally deleterious in nature, so lineages with more SVs could have suffered more deleterious mutations than those with less SVs, hindering their geographical spread[28]. Alternatively, the higher number of SVs in specialist sub-lineages could reflect the process of host-population adaptation via reductive genomic evolution[29].

To identify SV hotspots, we scanned the genome using 50,000 bp windows with 5000 bp steps and calculated fold-enrichment relative to the average SV density. Results indicated an uneven SV distribution, with a prominent hotspot between 3,200,000–3,600,000 bp enriched in *PE/PPE* genes (Fig. 2d). We found that 75% of SV breakpoints fall in genes. This is lower than expected by chance (mean random breakpoints = 92%, absolute *z*-score = 6.42, *p* value = 1e-10), likely due to the deleterious effect of SVs (Supplementary Fig. 11). Moreover, SV breakpoints were significantly associated with GC-rich sequences compared to randomly sampled genome regions (two-sided Wilcoxon test, *p* = 5e-8; Supplementary Fig. 12). *PE/PPE* genes are known for their GC-rich repetitive regions, which serve as an SV hotspot. These findings highlight that SVs in *Mtb* are not randomly distributed but are enriched in specific genomic features and regions.

## Pangenomic insights into structural variation in the ESX loci of *M. tuberculosis*

The *Mtb* genome encodes five type-7 secretion systems, which are known to play an important role in pathogenesis and virulence, the most studied being ESX-1[30,31]. ESX-5 is important for full virulence[32] and has been involved with various functions, such as the secretion of *PE/PPE* proteins, the activation of the host cell inflammasome response or macrophage death induction[31]. Its paralogs' (ESX-5a, ESX-5b, and ESX-5c) functions, however, are less understood. To determine the presence of SVs in the ESX-5 loci and which are more conserved, we analyzed the haplotypes from the 821 long-read assemblies mapped to the *Mtb*-PRG.

All ESX-5 loci had one SV bubble spanning their locus (Fig. 3a). Most isolates lacked SVs across these loci, with most non-reference haplotypes being present solely in 1–6 isolates. Nevertheless, the most frequent non-reference haplotype was a *ppe25-ppe27* deletion, present in 9% of all isolates (Fig. 3b). The deletion was found in all sub-lineage 4.4 isolates and sporadically in other lineages (L4.1, L4.3, L4.5, L1, L3, and L8). The deletion can most likely be attributed to non-allelic homologous recombination, as the 3' sequences of *ppe25* and *ppe27* are highly homologous (Supplementary Fig. 13), that is, the deletion spans the ppe25-ppe27 locus and removes *ppe25*, *pe18*, and *ppe26* while leaving *ppe27* intact. The deletion was found to provide higher persistence in mouse models and showed convergent evolution in *M. canettii*[33], which could explain its homoplasic effect across the *Mtb* phylogeny (Supplementary Fig. 14). A previous study documenting this deletion hypothesized that the loss of immunomodulatory *PE/PPE* genes could help the bacteria escape the immune system[34].

We extended the analysis to assess the impact of SVs across all of the ESX genes. To understand the probability of observing SVs in a gene, we calculated the trimmed (10%; removal of outlier genes with high SV counts) mean number of SVs in essential (0) and non-essential (0.76) genes[35,36], with 33% of genes being overlapped by an SV. Genes prefixed with the letters "ecc" and "mycP" in the ESX loci encode

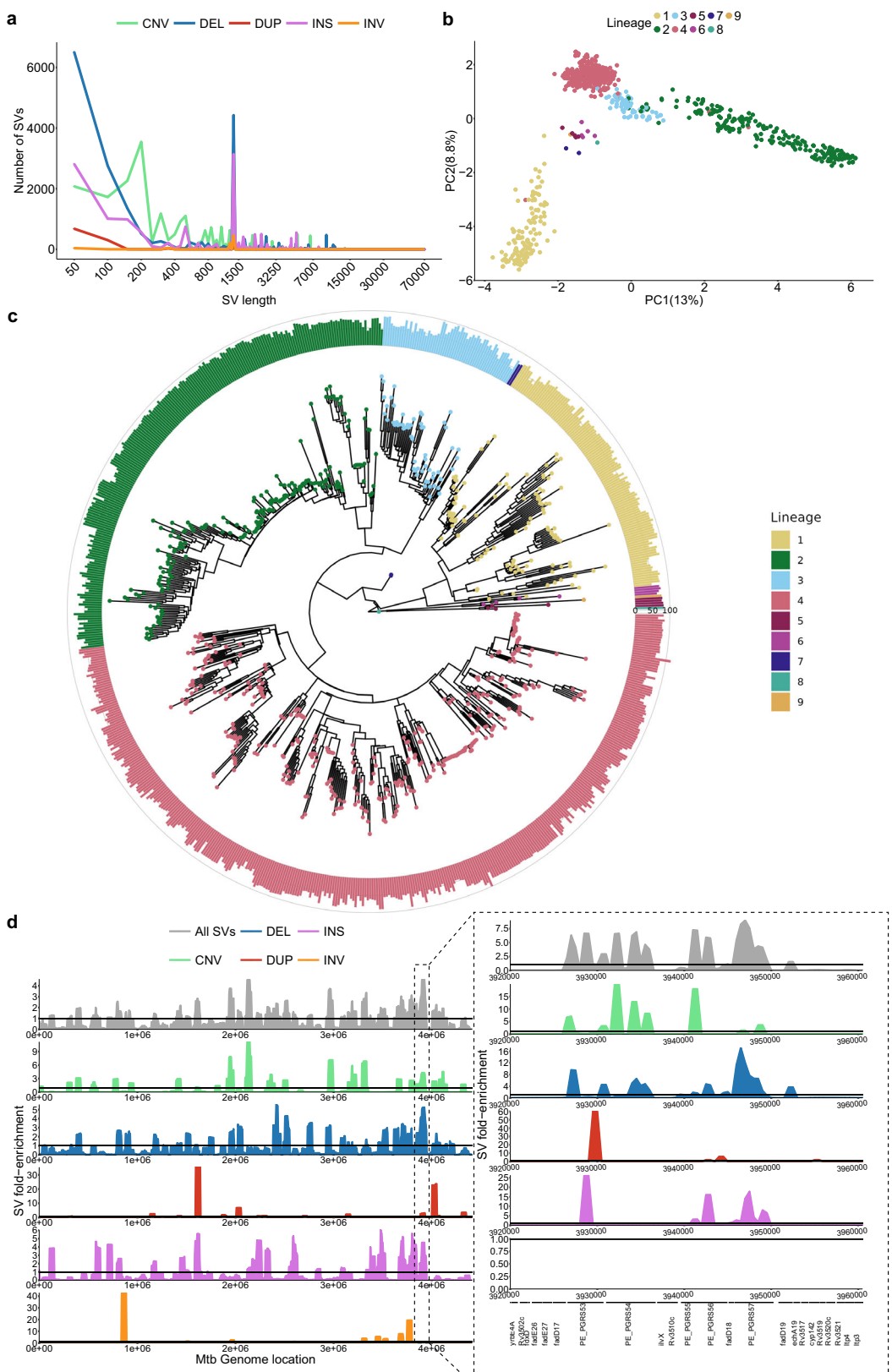

**Fig. 2 | Structural variation genotyped from 821 long-read *Mtb* assemblies covering L1–L9. a** Distribution of SVs in isolates by size and type. The x-axis is in log10 scale. **b** PCA based on the isolates' SV genotypes. **c** Phylogenetic tree filtering out 75 isolates with disconcordant SNP genotypes and lineage calls due to lower quality long-read SNP calls. Barplots represent the number of SVs compared to the ancestral reference sequence MTBC0. Rings, inner to outer, represent 0, 50, and 100 SVs, respectively. **d** Distribution of SVs along the genome by their fold-enrichment and type across all isolates. Horizontal lines represent no fold change of a specific position from the rest of the genome. The panel on the right represents the zoomed-in region of 3,200,000–3,600,000 with its own fold-enrichment values and genes along the region. Source data are provided as a Source Data file.

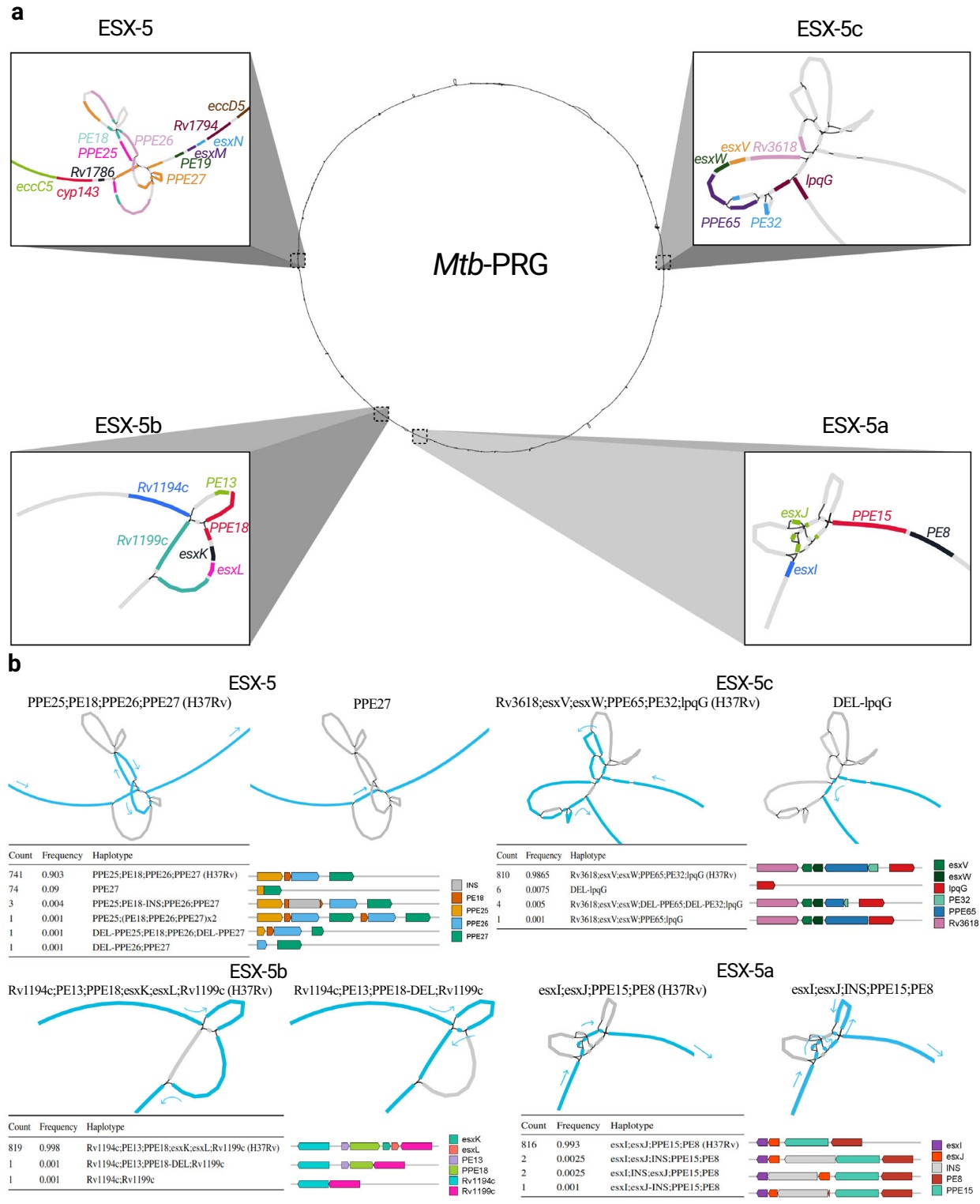

**Fig. 3 | Visualization of the structural variation in the ESX-5 loci through the pangenome reference graph. a** Locations of ESX-5, ESX-5a, ESX-5b, and ESX-5c in the *Mtb*-PRG. Different colors in each loci represent different genes. **b** Structural haplotypes of each ESX-5 locus. The order of the loci is the same as in (**a**), the graph representation is of the H37Rv reference on the left and of the non-reference most frequent haplotype on the right, for each locus. Nodes traversed through each haplotype are colored in blue with arrows pointing towards the direction of the genome. Tables and gene representations show the frequency of each haplotype and its effect on the gene content. Visualizations done using Bandage[98].

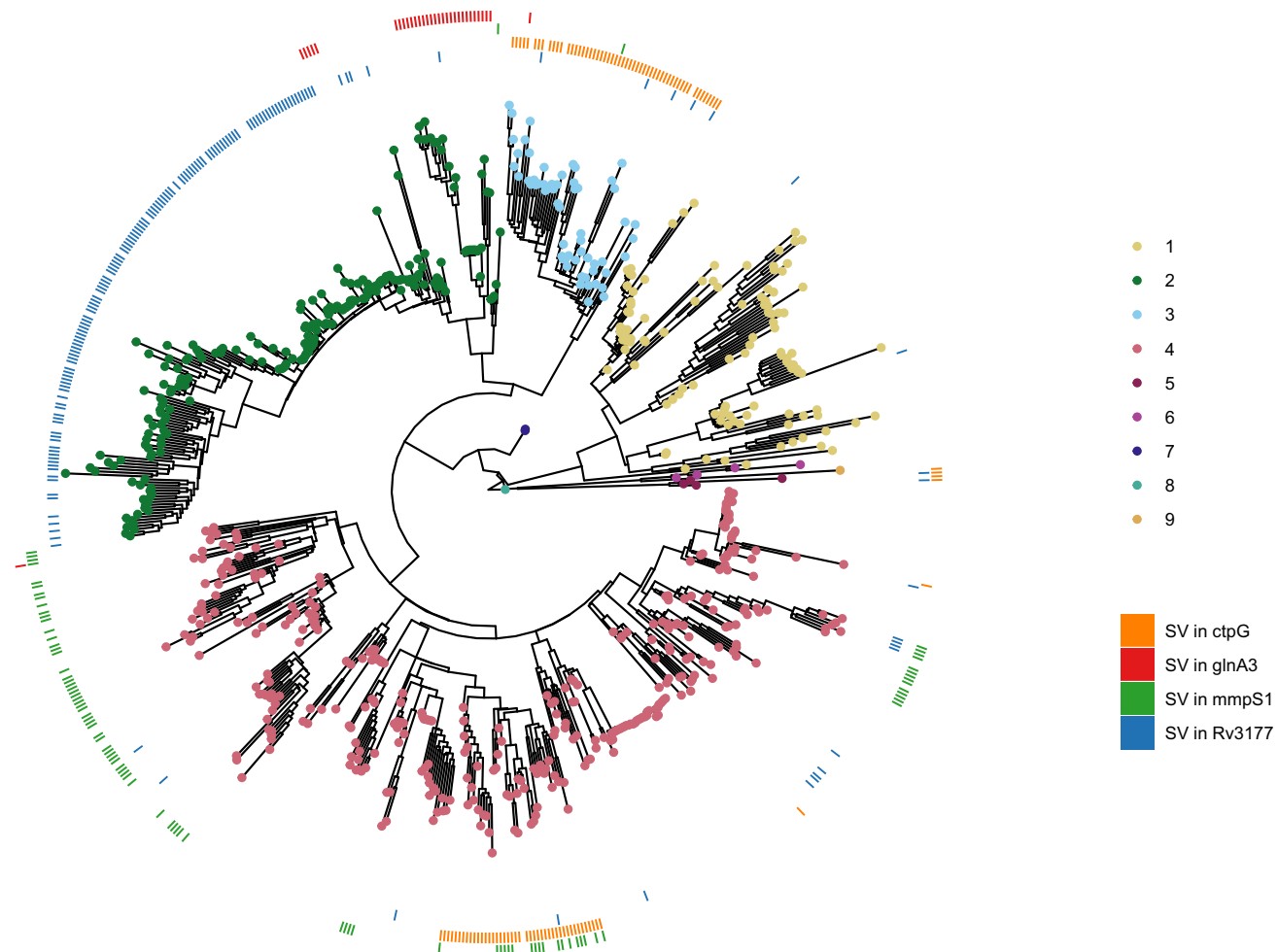

**Fig. 4 | Homoplasic effect of SVs overlapping genes across the phylogeny.** Same phylogenetic tree as in Fig. 2c. The outer rings represent the presence (color) or absence (white) of an SV in a specific gene for each isolate. The patchy distribution of some SVs within clades may be explained by inaccurate placement of isolates in the phylogeny, likely due to higher SNP error rates associated with long-read sequencing, especially in isolates generated using earlier versions of these technologies. Source data are provided as a Source Data file.

proteins which form the secretion system responsible for translocating proteins involved in virulence[37]. Consistent with their essential role in virulence[38], we found 0 isolates from our collection of 821 to have SVs impacting structural components of the ESX-3 and ESX-5 loci, and only one with an SV impacting ESX-1 genes *eccE1* and *mycP1*. In contrast, the ESX-2 and ESX-4 loci, which are non-essential and have yet to be ascribed a function[38], had between 1 and 16 isolates with SVs overlapping their gene members[38] (Supplementary Fig. 15a). Among all genes, *espI* from ESX-1 had the highest number of SVs, with 125 isolates exhibiting overlaps (Supplementary Fig. 15b). EspI acts as a negative regulator of the ESX-1 secretion system; thus, the high frequency of SVs in this gene may suggest positive selection for maintaining constant expression of the system[39]. Genes within the ESX loci prefixed with the letters "esx" encode proteins that are secreted through the ESX apparatus and thought to be implicated in host-pathogen interaction. Across our collection of 821 isolates, no *esx* genes in the ESX-1 and ESX-5 loci were found to have any SVs (Supplementary Fig. 15c), however *esx* genes in the ESX-2 locus displayed SVs in three isolates. Of the ESX-5 paralogs, ESX-5a was the most strongly conserved (1 isolate with an SV falling in a gene), suggesting deletions of this locus are less well tolerated than deletions in ESX-5b and ESX-5c (Fig. 3b and Supplementary Fig. 15c, d). Interestingly, one instance of a full ESX-5c deletion was found in the geographically restricted African lineages (L6 and L9).

## A genome and phylogeny-wide screen of SV positive selection in *M. tuberculosis*

We performed a systematic screen for genes which evolved SVs repeatedly across the phylogeny, following the rationale that this may reveal genes targeted by positive selection[40]. The criteria for our screen stipulated that the gene had to be affected by an SV in at least ten isolates, belonging to more than one *Mtb* lineage. We further excluded genes in or ±3000 bp of low-mappability regions[10], as those regions are more prone to structural rearrangements, which could be falsely perceived as positive selection. This analysis resulted in 59 homoplasic gene candidates. The gene with the strongest homoplasic effect was *Rv3177*, with SVs in 158 isolates from L1.2.1, L1.1.1, L2.2.1 (82% of all isolates), L2.2.2, L3.1.2, L3.1.3, various L4 sub-lineages and L5 in 15 different breakpoints across the gene (Fig. 4). *Rv3177* is a probable peroxidase involved in detoxification of reactive oxygen species (ROS). In a previous study, variants in *Rv3177* were associated with isolates harboring the isoniazid resistance-conferring *katG*-S315T mutation, suggesting a possible compensatory role in drug resistance[41]. Another gene that acquired SVs in different branches of the phylogeny was *ctpG* with six different DEL breakpoints across the gene (Supplementary Fig. 16) in L3, L4.3.4 and L5 (Fig. 4). CtpG is a zinc (Zn) exporter whose deletion has been associated with Zn accumulation[42]. During host infection, *Mtb* goes from Zn-rich to Zn-limited environments[43]. A non-functional Zn exporter could possibly provide bacteria in Zn-limited

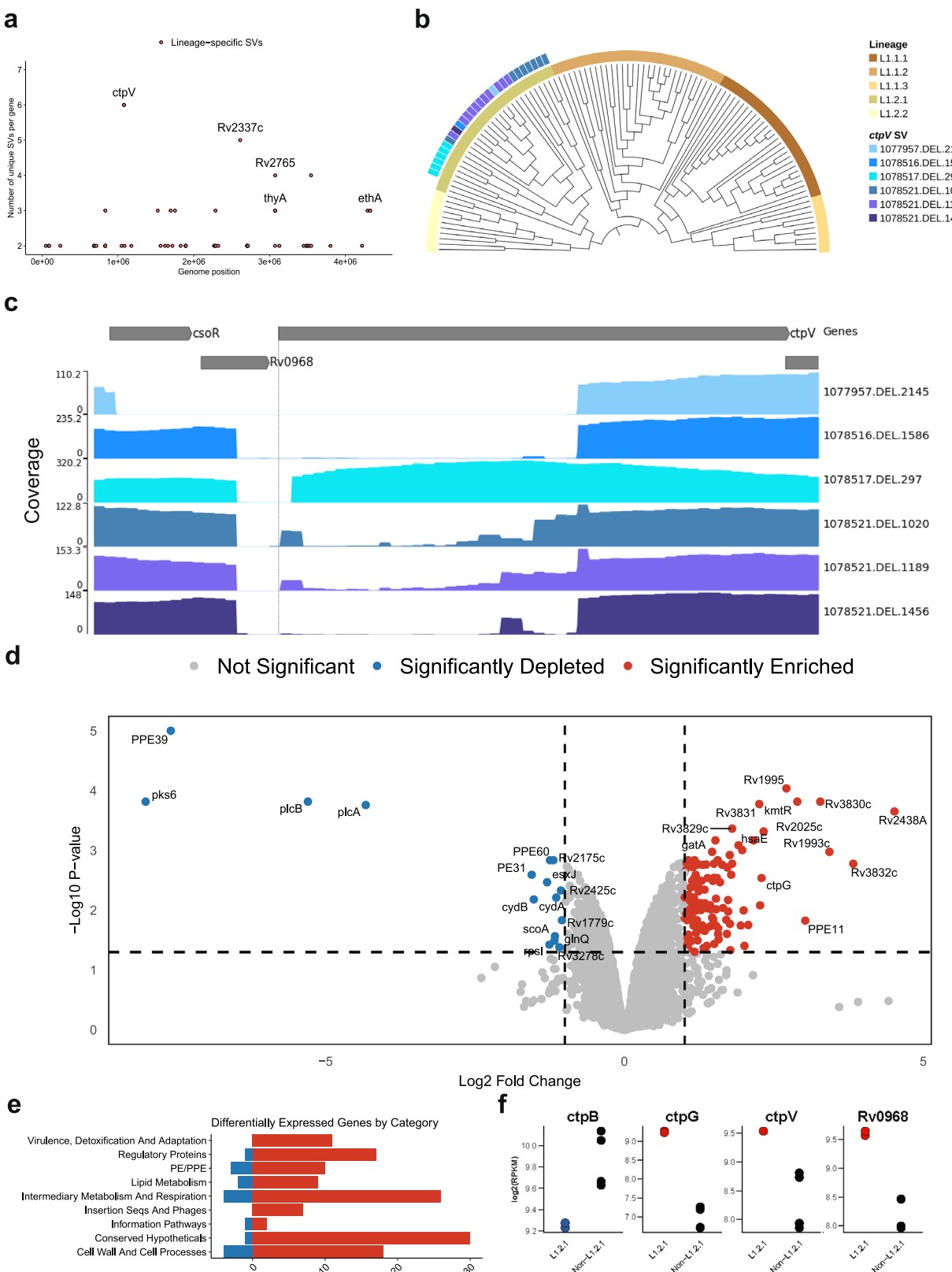

conditions a selective advantage. Other interesting genes were *mmpS1*, with SVs in various L4 sub-lineages with a prominence in L4.1 as well as a few isolates in L2 and L3, and *glnA3*, with SVs in all L2.2.2 isolates and sporadically in L2.2.1, L3, and L4.1 (Fig. 4).

We sought to validate these results within a larger cohort. To do so, we downloaded a list of 44,709 high-quality Illumina sequences

grouped in a previous study[17]. After assembly, we ended with 41,134 SV VCF files. We found a few isolates with SVs in *Rv3177* and *mmpS1*, as *Rv3177* is in a large bubble, likely hard to genotype using short-read data, and the *mmpS1* SVs were all INSs, variants with low genotyping recall using short-read data. We were able to confirm the same distribution of SVs for *ctpG* and *glnA3* (Supplementary Fig. 17). A similar

**Fig. 5 | Discovery of growing *ctpV* deletions across the L1.2.1 phylogeny and a transcriptomics analysis. a** All genes across the genome with the number of isolates that have SVs overlapping them on the y-axis. Red-colored points represent those that passed the second positive selection screen; i.e., those genes that have different SVs in the same gene, unique to a lineage. **b** Phylogenetic tree of all lineage 1 isolates, showing those that present *ctpV* deletions and/or non-synonymous mutations. SV IDs are formulated as following "{start position of SV}.{SV type}.{size of SV}". Tree visualized with TreeViewer[99]. **c** Read coverage across the *csoR* operon for each unique deletion spanning *ctpV*. Y-axis shows the read depth at each genomic position. SV ID is the same as (**b**). Visualization done using pyGenome-Tracks. **d** Volcano plot summarizing the differential expression results comparing L1.2.1 against L1.1.2 and L1.1.1 isolates. *P* values are calculated using a two-sided *t*-test, with Benjamin–Hochberg multiple hypothesis correction. Log fold change on the x-axis. **e** Functional categories from Mycobrowser of the significantly differentially expressed genes. **f** Dot plots showing expression patterns for four copper-associated genes in L1.2.1 against L1.1.2 and L1.1.1. In red are genes enriched, and in blue are genes depleted in L1.2.1. Source data are provided as a Source Data file.

distribution was observed for SVs in *ethA*, a gene whose deletion is associated with ethionamide (ETO) resistance, with a main cluster located in the Beijing lineage. For *pncA*, a pyrazinamide (PZA) resistance-associated gene, we found less isolates with SVs and the distribution was not as clustered, though mainly isolates in L2 had SVs in that gene. These results showcase the added benefit of using a pangenome to identify key genes implicated in *Mtb* evolution, and highlight four candidates for prioritization in future research.

## Characterization of a *M. tuberculosis* lineage 1.2.1-specific copper exporter deletion

We expanded the previous screen to detect genes with multiple lineage-specific SVs. This analysis revealed diverse deletions of the *ctpV* gene, unique to the L1.2.1 sub-lineage (excluding its deepest branch split; Fig. 5a and Supplementary Fig. 18). These deletions increased in size further from the branch split (Fig. 5b) and overlapped the gene's start codon, indicating a loss-of-function mutation (Fig. 5c). SNPs could not be reliably called for all the isolates as long-read data were used, nevertheless, we found no loss-of-function variants in this gene in other L1 sub-lineages. CtpV is a putative P-type ATPase copper (Cu) exporter protein, with the main Cu binding site located in the first 66 amino acids. A multiple sequence alignment of the protein sequence among diverse mycobacteria showed >80% pairwise identity across the *Mycobacterium* genus (Supplementary Fig. 19), aside from *Mycobacterium leprae*, highlighting the importance of this protein not only in *Mtb*.

L1.2.1 is predominant in Southeast Asia, especially the Philippines, the fourth country in the world with the highest burden of active tuberculosis[1], accounting for ~65–80% of all lineages in the country[44,45]. Moreover, this sub-lineage shows high drug resistance rates, high transmissibility and is the most prevalent L1 sub-lineage globally[46,47]. Given these characteristics, we hypothesized that this deletion impacts the expression of copper-associated genes, potentially contributing to the unique adaptive traits of this sub-lineage. To determine this, we compared the transcriptional profile of an L1.2.1 isolate (297bp *ctpV* confirmed deletion) against L1.1.1 and L1.1.2 isolates under normal growth conditions[48] (Supplementary Table 2 and Supplementary Fig. 20). This analysis revealed 147 differentially expressed genes (Fig. 5d). Functional categorization revealed that the L1.2.1 isolate showed enrichment of genes involved in virulence, detoxification and adaptation (Fig. 5e). *ctpG* and *Rv0968* were among the differentially expressed genes (Fig. 5f). *Rv0968* is part of the *cso* operon, where *ctpV* lies. This operon is controlled by CsoR, which inhibits its own expression in normal conditions. However, when Cu is present, it attaches to CsoR and inhibits its function, enabling *Rv0968* and *ctpV* expression[49,50]. The higher expression of both *Rv0968* and *ctpV* (non-significant; adjusted *p* value = 0.053) was indicative of Cu accumulation inside these isolates (Supplementary Fig. 21). Moreover, the enrichment of *ctpG*, another P-type ATPase metal transport protein, has been previously associated with Cu accumulation in the cell, as well as the depletion of *ctpB*, a P-type ATPase Cu transport protein associated with Cu uptake in Cu-replete medium (nonsignificant; adjusted *p* value = 0.059)[50,51]. Nevertheless, genes associated with oxidative stress (i.e., *sigE,furA,nuoB*) or protein stress (i.e., *rpsR,rplI,rpsQ*) were not differentially expressed, indicating that Cu was not present at toxic

levels (Supplementary Data 3)[50]. Altogether, these results are indicative of higher Cu accumulation in an L1.2.1 isolate compared to other L1 sub-lineages.

## Discovery of DR-associated SVs in 41,134 *M. tuberculosis* isolates

Using the *Mtb*-PRG and miniwalk, we aimed to discover DR-associated SVs across various first and second-line drugs for treating TB. The previous list of 41,134 isolates also contained phenotypic information for at least one drug, with first-line drugs being vastly more represented than second-line drugs (Fig. 6a). To search for associations, we used two parallel methods: Firth logistic regression and a supervised machine learning model (xgboost), testing each drug independently and accounting for population structure and TB-Profiler predictions[52]. Analyses were conducted at both the SV and gene levels (SV-gene burden test).

Logistic regression revealed 14 non-canonical SVs (post-filtering) significantly associated (adj. *p* value <0.05) with seven DR phenotypes (Supplementary Data 4). Associations in *PE/PPE* genes were excluded due to their SV hotspot nature (list of discovered associations: see Supplementary Data 5). Xgboost outputs the estimated importance of each SV and TB-Profiler prediction to the analysis (ranging between 0–100%), indicating the relevance of these factors in classifying DR phenotypes. This analysis highlighted the contribution of the current list of WHO mutations, as identified by TB-Profiler, in predicting DR, with first-line drugs as well as fluoroquinolones and aminoglycosides having high importance scores (importance >50%). However, newer drugs such as bedaquiline, delamanid or clofazimine showed lower importance scores (Fig. 6b). The SV-gene burden test revealed 27 non-canonical genes associated with drug resistance across ten drugs (Supplementary Data 6). To increase confidence in our findings, we performed, for each associated gene and SV, a permutation analysis to estimate the likelihood of our observed *p* values under a null distribution. This showed that under the null hypothesis, there's <5% chance of observing these associations (absolute *z*-score range 1.65–2.57), suggesting statistical significance for all variants.

For isoniazid (INH), we found two SVs and three genes associated with DR (Fig. 6c and Supplementary Figs. 22, 23). The most significant SV was a deletion overlapping *Rv3434c* (Fig. 6c), a gene previously associated with kanamycin (KAN) and amikacin (AMK) DR[53]. An overwhelming 99% of isolates with a mutation in this gene presented an isoniazid resistance phenotype, the vast majority from L2. Nevertheless, only five isolates did not present TB-Profiler predictions for INH DR. Another deletion spanning *Rv0738-Rv0741* was predominantly found in resistant isolates (97%), with an *Rv0740* knockout having previously been associated with INH resistance[54]. Five isolates with SVs in *Rv0740* did not have TB-Profiler predictions for INH DR. These findings underscore the potential of SVs in identifying additional resistance mechanisms, particularly in a widely distributed and commonly used drug like INH, not captured by TB-Profiler. For rifampicin, no non-canonical SVs or genes with SVs were detected in our analysis. For PZA, we found four SVs and six genes. The highest-scoring gene was *pncA*, with 2.3% of DR isolates having SVs overlapping that gene (Supplementary Fig. 23). An operon with SVs and also highly associated with PZA-DR was *Rv1505c-Rv1507c*, which plays

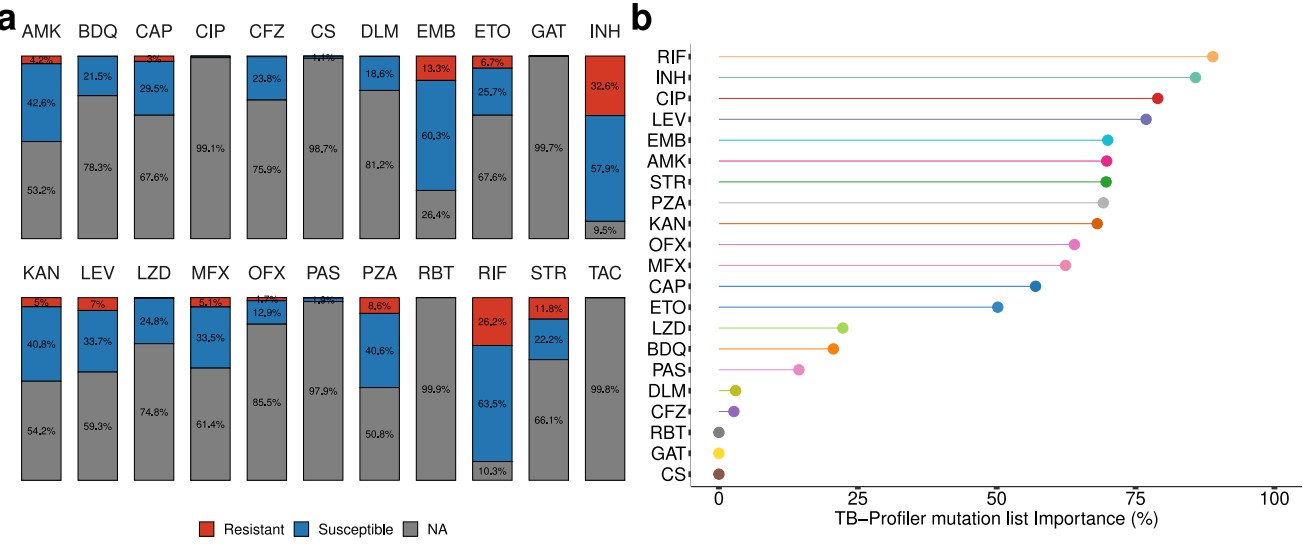

**Fig. 6 | Resistance-conferring SVs across 41,134 isolates and 22 drugs. a** Stacked barplot showing the percentage of isolates that are susceptible, resistant or have no information (NA) across the 41,134 isolates, for each drug. Percentage labels are only shown when the value is ≥1%. **b** Machine learning importance scores (%) of the TB-Profiler mutations predictions. TAC is not illustrated as only resistant isolates were available. **c** Thorough description of the 14 associated SVs, ranked by their adjusted *p* value (top - lower *p* value). AMK amikacin, BDQ bedaquiline, CAP capreomycin, CIP ciprofloxacin, CFZ clofazimine, CS cycloserine, DLM delamanid, EMB ethambutol, ETO ethionamide, GAT gatifloxacin, INH isoniazid, KAN kana-mycin, LEV levofloxacin, LZD linezolid, MFX moxifloxacin, OFX ofloxacin, PAS para-aminosalicylic acid, PZA pyrazinamide, RBT rifabutin, RIF rifampicin, STR strepto-mycin, TAC thioacetazone. Source data are provided as a Source Data file.

a role in diacyltrehalose biosynthesis, a lipid that forms part of the cell wall[55]. The only gene with SVs associated with ethambutol (EMB) DR was *accE5*, whose encoded protein is found in the cell wall[56] (Supplementary Fig. 23).

For second-line drugs, we found non-canonical SVs and genes with SVs for AMK, bedaquiline (BDQ), capreomycin (CAP), cipro-floxacin (CIP), clofazimine (CFZ), delamanid (DLM), ETO and

streptomycin (STR) (Fig. 6c, Supplementary Fig. 23, and Supplemen-tary Note 2). Our results highlight the importance of different types of SVs in DR, such as an insertion associated with STR DR, as well as the role of non-coding regions, such as a deletion upstream *Rv2258c* associated with CAP DR. The low importance scores of second-line, recently introduced drugs (BDQ, CFZ, and DLM) highlight the need to find more DR variants.

## Discussion

SVs have a higher potential to disrupt gene-coding and cis-regulatory regions than SNPs due to their larger sizes, yet they have been excluded and understudied in many species[9,57]. Their role in *Mtb* pathogenesis has been vaguely explored, mainly in uncovering their role in drug resistance[7,11,14] and evolution[6,58]. Nevertheless, previous studies have focused on specific genes, had small sample sizes lacking lineage diversity, were limited to first-line drugs or used short-read data, reducing the potential to uncover the role of these variants. In this study, we leveraged both long-read sequencing data and genome graph methodologies across thousands of isolates to reveal how SVs affect diverse aspects of *Mtb* pathogenesis.

To profile the diversity and effect of SVs across the *Mtb* genome, we constructed a *Mtb*-PRG from 859 long-read assemblies. The inclusion of at least one isolate across lineages 1–9 in the *Mtb*-PRG was important to capture genomic diversity. Miniwalk was developed to use the output from minigraph to genotype SVs against a reference genome. Benchmarking showed that using minigraph and miniwalk on long-read assemblies performs similarly to SVIM-ASM. Using minigraph and miniwalk on short-read assemblies provided a significantly higher precision than using a linear genome and manta. The precision increase is seen especially in insertions and large deletions. Recall is not significantly higher as minigraph provides an NA whenever it cannot correctly map the assembly through a bubble. As miniwalk complements minigraph's output, it could be used with graphs from other species, allowing for wider applications. In particular, while miniwalk's performance on multi-chromosome bacteria or plasmid-rich species remains untested, we expect it to be similarly effective in these contexts, as minigraph is capable of handling multiple chromosomes. However, given that this study did not evaluate multi-chromosome or plasmid-rich genomes, we recommend applying miniwalk primarily to single-chromosome genomes at present, with extension to multi-replicon contexts left for future work. One current limitation of miniwalk would be that it needs assemblies or long reads in order to genotype SVs. A future direction for miniwalk lies in genotyping SVs using unassembled short reads, which could also improve the recall. Our findings illustrate the advantages of adopting pangenome-based approaches for SV detection, highlighting miniwalk as a useful resource.

We provide a description of 3077 SVs in the *Mtb* genome across 821 diverse isolates with long-read data. When investigating the structural variability of the ESX loci, we revealed that ESX-1, -3, and -5 are more conserved than ESX-2 and -4, and that ESX-5a is the most conserved out of the three ESX-5 paralogs. This suggests that the function of ESX-5a could be more essential for the bacterium. ESX-5c, on the other hand, was found to be completely deleted in all L6 and L9 isolates, showing its redundancy regarding bacterial viability. A frequent *ppe25-ppe27* deletion was observed in all sub-lineage 4.4 isolates, a variant previously associated with greater bacterial persistence in a mouse model[33]. Interestingly, the gene with the most SVs in the ESX loci across all isolates was *espI* (15% of all isolates), from the ESX-1 locus. *espI* is located in a region of low-mappability, making it challenging to determine whether these SVs are under positive selection. Nevertheless, a previous study[39] found that, in the presence of BDQ, ESX-1 secretion is blocked due to ATP repletion, yet this is corrected with a loss of *espI*. These findings warrant further investigation of *espI* mutations in clinical DR isolates using long-read data.

L1.2.1 is the most prevalent L1 sub-lineage globally, with L1.2.1.1.1.2 being geographically unrestricted while L1.1.1.1 and L1.1.3 are geographically restricted[47]. L1.2.1 predominantly affects individuals in the Philippines and Indonesia, the two countries with the highest rise in TB incidence from 2020 to 2023[1]. Understanding the higher transmissibility of this sub-lineage could reveal *Mtb* mechanisms enhancing bacterial survival or host infection efficiency, which could have a direct implication for TB surveillance in these regions. Moreover, knowledge of transmissibility can inform better designs on sub-lineage-specific

vaccine candidates[59]. A lineage-specific SV screen revealed L1.2.1-specific *ctpV* (a copper exporter transmembrane protein) deletions. The transcriptional profile of an L1.2.1 isolate with the deletion against other L1 sub-lineages revealed enriched genes associated with copper accumulation, though not toxicity. Among other effects, copper can inhibit PdtaS, an important transcriptional repressor, which could explain the enrichment of genes associated with virulence and detoxification in this sub-lineage[60]. However, more evidence is still needed to directly link the *ctpV* deletion to Cu accumulation. Additionally, *Mtb* has other copper exporter proteins[61], which would explain the viability of these lineages despite having the *ctpV* deletion. It remains to be seen whether other variants occur on the branch housing the *ctpV* deletion, and whether they contribute to the virulence of the sub-lineage. A limitation of the supporting RNA-Seq data is that it was generated in only one isolate for each sub-lineage, which may not represent the transcriptional profile of all isolates in those sub-lineages. To overcome this and directly test the functional consequences of the *ctpV* deletion, future studies should include intracellular copper quantification, copper-stress growth assays, and genetic complementation experiments. Additionally, animal infection models that support granulomatous lesions, where copper levels are elevated under hypoxic conditions[61], could provide further insight into whether this deletion confers an adaptive advantage in vivo. Regardless, our study identifies a gene that may contribute to the higher transmissibility of L1.2.1. Other positive selection signatures across lineages revealed deletions in another metal exporter, *ctpG*, highlighting the importance of metal homeostasis in *Mtb* adaptation to different host environments.

By accounting for WHOv2-listed mutations, we were able to detect non-canonical SVs, and genes with SVs, associated with DR across 11 drugs, including important, recently introduced second-line drugs. Consistent with previous studies[53,62], the identified SVs affect both coding and intergenic regions, emphasizing the importance of the non-coding regions of the genome. Our graph approach facilitates the discovery of these SVs, which could be especially beneficial in those settings where whole-genome sequencing is routinely done to diagnose DR-TB[2,3]. The high frequency of SVs in ETO- and PZA-DR isolates showcases the need to improve SV detection in clinical strains. Since our approach relies on SVs found in 859 assemblies, future studies that focus on long-read sequencing isolates with phenotypic DR might add undiscovered DR-associated SVs to the *Mtb*-PRG. As recent drugs were underrepresented in our DR Illumina dataset, the lack of power might not have allowed us to detect important SVs associated with DR; as more Illumina data becomes available for these drugs, the power to detect associations will improve. We hypothesize that the lack of DR-unique SVs in our results may reflect the limitations of binary resistance labels, as many SVs may influence minimum inhibitory concentration (MIC) without conferring full resistance. As shown in recent studies[63,64], using MIC instead of binary phenotypes might enhance low-level DR variants detection. Finally, external validation of the associations should be done to further confirm our findings.

The *Mtb*-PRG and miniwalk tools developed here, which are now openly accessible to the TB research community, open new opportunities to investigate the contribution of SVs to drug resistance, evolution and *Mtb* pathogenesis.

## Methods

### Public *Mtb* long-read sequence and assembly datasets

We gathered available long-read (Pacific Biosciences, Oxford Nanopore Technology) sequence data and assemblies from public repositories using fastq-dl (v2.0.4)[65] (relevant ENA/SRA/GCA IDs to access all data can be found in Supplementary Data 1).

### *Mtb* culture and DNA extraction

Genomic DNA was extracted from *Mtb* isolates grown on solid culture utilizing the FastPrep 24 homogenizer, followed by purification and

concentration of the crude cell lysate using the QIAamp 96 DNA QIAcube HT Kit.

## Long-read sequencing of underrepresented *Mtb* isolates

Sequencing was done in two batches for African (L5, L6, and L9) and non-African lineages separately. For African lineages, DNA was quantified using the Quant-iT 1X dsDNA high sensitivity assay kit (Q33232) (Thermo Fisher Scientific) as per the manufacturer's instructions. Library preparation was performed using the Rapid Barcoding Kit (SQK-RBK004) (Oxford Nanopore Technologies), as per the manufacturer's instructions. Briefly, DNA (variable input ranging from 60 to 180 ng) was fragmented enzymatically and barcode adapters ligated to each sample. The barcoded libraries were then pooled and sequencing adapters added prior to loading onto an R9.4.1 flow cell. Libraries were sequenced on the GridION Mk1 platform (Oxford Nanopore Technologies). Basecalling was done using Guppy (v7.2.13) and the "dna_r9.4.1_e8_hac@v3.3" model.

For non-African lineages, sequencing was performed using the Ligation Sequencing Kit (SQK-LSK109) and the Native Barcoding Kit 1D (EXP-NBD104 and EXP-NBD114) according to the manufacturer's instructions on the GridION platform with R10.4.1 flow cells. Basecalling was done using dorado (v0.5.3) and the "dna_r10.4.1_e8.2_400bps_sup@v4.3.0".

## Sequencing data quality control, assembling, NCBI assemblies, and assembly quality control

Nanopore raw fastq files were preprocessed using Porechop (v0.2.4)[66] to remove all adapters in the sequences and nanoq (v0.10.0)[67] to filter out reads that were under 500 bp in length and with a quality lower than 7. PacBio CLR fastq files were preprocessed with seqkit (v2.3.1)[68] by removing reads shorter than 500 bp and duplicate reads. Pbclip[69] was used to correct reads in case the PacBio toolchains handled the raw to fastq file conversion improperly. For Pacbio-Hifi reads, HifiAdapterFilt (v2.0.1)[70] was applied to detect and remove possible adapters that were left in the samples and seqkit was applied to remove reads shorter than 500 bp. After read preprocessing, reads from each technology were decontaminated as described in ref. [71], which, in short, is a two-step process that uses kraken2 (v2.1.2)[72] to firstly extract human reads from a sample and after that selects solely those reads classified as belonging to *Mtb*. Flye (v2.9.2)[18] was used to create the assemblies. Assemblies were finally polished with their own long reads to correct for assembling mistakes using Pilon (v1.23)[73].

Long-read-sequenced NCBI assemblies were also downloaded to obtain a wider dataset and, jointly with the long-read assemblies, were run through QUAST (v5.2.0)[19]. A threshold of 95% was used for completeness to filter incomplete assemblies. Contiguity was assessed by applying an N50 filter; the largest repetitive sequence in the *Mtb* genome (10,000 bp). Correctness was assessed by applying a number of misassembly filters of a maximum of 100. While this threshold excluded only 24 assemblies, all of them generated with PacBio CLR (non-HiFi) data, we chose to exclude all PacBio CLR assemblies from the analysis due to their consistently poor quality across multiple metrics. To evaluate potential bias introduced by the misassembly filter, we examined the relationship between k-mer-based distance to H37Rv (via Mash (v2.3)[74]) and QUAST misassembly counts. We observed a negative correlation ($p$ value = 7e-9; Supplementary Fig. 24), indicating that more divergent strains were not disproportionately affected by the misassembly threshold. Contigs with a length shorter than 1000 bp were removed using sequence-stats (v1.1)[75]. Removing assemblies according to their length was not necessary, as the remaining ones were all inside the reference genome length range by 10% (4,852,685.2– 3,970,378.8). Moreover, filtering by the number of contigs was not applied as the maximum number of contigs an assembly had was 589, far from the filter applied by ref. [76] (2000).

## Lineage assignment

Long reads used to create the assemblies were sent through TB-Profiler (v5.0.1)[52] which assigned the lineages to each sample from the fastq data using[4]'s lineage nomenclature. For the NCBI assemblies, as they weren't in fastq format, they were mapped to the reference genome first using minimap2's -x asm5 flag (v2.24)[77] as well as its paftools.js script to transform the alignment into a VCF file with the detected SNPs. The file was then sent to fast-lineage-caller (v0.3.2)[47] to determine the lineage under the same scheme as the long-read assemblies. Isolates that were assigned more than one lineage, a *Mycobacterium bovis* lineage or an N/A were filtered out.

## Pangenome reference graph construction

To create the PRG, minigraph (v0.20-r559)[20] was used with -l 10,000 -d 5000 parameters. The backbone of the PRG was the reference genome H37Rv, and the rest of 859 samples were added to the PRG. After running minigraph with the -cxggs flag, the PRG was created. Nodes containing long mononucleotide sequences were removed from the dataset, as these are likely artifacts resulting from sequencing errors in long-read technologies. Such errors commonly occur in homopolymer regions, where the sequencer struggles to accurately determine the number of repeated bases.

## Genotyping SVs using miniwalk

When mapping assemblies to minigraph using default parameters, highly fragmented assemblies are filtered out, however, this can be overcome by applying the -l 10,000 -d 5000 flags. Nevertheless, to find robust SV signals across less fragmented assemblies, we used default parameters and filtered out 38 assemblies. SVs of 821 assemblies were genotyped using our tool, miniwalk (v0.1). Genotyping is done in six steps. First, we create an assembly, next we map it to the PRG using minigraph. Following that, our assembly's path is merged with a reference path (in this case, H37Rv's path through the PRG) to create a VCF. The VCF is then refined using miniwalk in the last three steps: using mod to output specific SVs and their positions, ref to order the SVs and cluster them together where necessary and ins2dup to change assigned INSs to DUPs where necessary (Supplementary Methods 2).

## Creating an *Mtb* SV gold-standard

As there exists no *Mtb* SV gold-standard to benchmark against, we used a set of high-quality *Mtb* assemblies[21] and mapped them to the reference genome H37Rv using minimap2 with the -x asm5 option and SVs were called using SVIM-ASM (v1.0.3)[22] in haploid mode. The resulting VCF would be used as the gold-standard.

## Calling SVs on short-read Illumina data

In this work, we called SVs on Illumina data using two different methods. We did quality control on the Illumina reads using fastp (v0.23.2)[78] and filtered out human reads using kraken2, mapped them to H37Rv using bwa mem (v0.7.17)[79] and called SVs using manta (v1.6.0)[23]. SVs in the resulting VCF file that were marked as "BND" were filtered out. For the second method, we used the filtered reads to create an assembly using shovill (v1.1.0)[25], mapped the assembly to the *Mtb*-PRG using minigraph and called SVs using miniwalk.

## Benchmarking SVs called mapping to H37Rv against *Mtb*-PRG and miniwalk

To evaluate the SVs called using minigraph and miniwalk, we did three separate benchmarks, calculating the precision and recall. The first benchmark consisted of comparing the SV calls of polished long-read assemblies with minigraph + miniwalk against H37Rv + SVIM-ASM as the truth. SVIM-ASM was chosen as it is a well-established assembly SV caller reliant on a linear genome, which is what differentiates it from minigraph/miniwalk. The second consisted of comparing the SV calls

of short-read Illumina data with minigraph + miniwalk or H37Rv + manta against H37Rv + SVIM-ASM as the truth. The third benchmark consisted of comparing the SV calls of short-read Illumina data with minigraph + miniwalk or H37Rv + manta against minigraph + miniwalk (polished long-read assembly SVs) as the truth. We used miniwalk's bench mode to calculate precision and recall. Precision was defined as the rate of true positives (SVs correctly called when comparing to a truth set) over the total sum of true positives and false positives (SVs called but not in the truth set) (TP/(TP + FP)) while recall was defined as the rate of true positives over the total sum of true positives and false negatives (SVs that were not called but were in the truth set) (TP/(TP + FN)). A true positive was determined as such in the case where the called SV overlapped ≥25% of a truth SV[80], matched the sequence at least 50% with a truth SV, in case both were in the same tandem duplication[81] or spanned two standard SVs that were <200 bp apart or vice versa (Supplementary Methods 2). Miniwalk's **bench** code was tested by using synthetic VCF files with expected precision and recall values across different SV representations (Supplementary Methods 2).

For simulation benchmarking, we used vcflib[82] to introduce SVs from a real assembly into the H37Rv reference, then simulated a long-read assembly for evaluation (Supplementary Methods 3). This setup provides a controlled test with known ground truth while maintaining realistic SV profiles.

The first and second benchmarks were repeated using an external tool, vcfdist (v2.3.2)[26], to validate the results found using miniwalk bench. The VCF file resulting from minigraph + miniwalk was slightly modified to be accepted by vcfdist, such as replacing the asterisk where there is no sequence in the REF or ALT columns for the corresponding base in that position. Vcfdist was then run with -ct 0.001 to determine the ct score for all SVs.

## L9 translocation detection

The largest SV identified in our dataset was a ~66,000 bp rearrangement in an L9 isolate, involving both an inversion and a translocation into a novel genomic position. Since minigraph does not natively represent translocations in the graph structure, such events are fragmented across non-adjacent nodes. As a result, miniwalk reported this event as two separate SVs of similar size—an INS and a DEL—without resolving the rearrangement as a translocation (Supplementary Fig. 25).

To investigate the true nature of this event, we aligned the L9 long reads to the H37Rv reference using minimap2. Manual inspection in IGV revealed that reads mapping to the insertion breakpoint also aligned to the distant deletion site, consistent with a translocation. Furthermore, reads spanning the region corresponding to the deleted sequence were mapped in the reverse orientation, indicating that the translocated segment was also inverted in its new genomic location.

## PRG assemblies SV PCA

SVs that were ±25bp of proximity, ±50 bp in size and the same type were considered as the same SV across different isolates. A genotype matrix was then created with isolates as rows and all found and clustered SVs as columns, and a PCA was done using the factoextra package (v1.0.7) in R (v4.1.2).

## PRG *Mtb* phylogenetic tree

To construct a phylogenetic tree, the SNPs of all isolates used to create the PRG was needed. For NCBI assemblies, SNPs were obtained as mentioned in "Lineage assignment". For long-read data, reads were mapped to H37Rv using minimap2, variants called using Pilon, non-SNP variants were excluded and SNPs of QUAL >100 were kept. VCF files were merged and consensus fasta files were created using a custom script. The phylogenetic tree was inferred using RAxML (v8.2)[83] using a GTR model of nucleotide substitution and the L8 isolate as the

outgroup. To improve the accuracy and interpretability of the SNP-based phylogeny, we excluded 75 assemblies that produced discordant lineage calls and phylogeny placement. This was likely the result of using earlier long-read technologies, which are associated with higher SNP calling error rates[84].

## Pairwise identity calculation

Pairwise sequence identities were calculated using a custom Python script implemented with Biopython and NumPy. A multiple sequence alignment of the CtpV protein amino acid sequences across various *Mycobacterium* species was first loaded from a Clustal-formatted file. For each pair of sequences in the alignment, a pairwise identity score was computed by comparing aligned positions and calculating the proportion of identical residues.

## L1.2.1 transcriptome analysis

Public Illumina RNA-Seq data were downloaded using fastq-dl. All L1 isolate data from ref. 48 was downloaded; RNA-Seq data from L1.2.1, L1.1.1, and L1.1.2 in normal growth conditions, with two replicates per isolate. Reads were preprocessed using fastp and mapped to the reference genome H37Rv with bwa mem. To identify the strand specificity of the RNA-Seq libraries, the mapped reads were visualized in Integrative Genomics Viewer (IGV)[85]. Read duplicates were removed using samtools (v1.16.1). Counts of reads mapped to each gene were obtained using featurecounts (v2.0.8)[86]. To quantify the transcriptional expression, we followed the preprocessing steps of ref. 87. We removed small genes (≤150bp), non-coding transcripts (tRNA, rRNA, and annotated non-coding RNAs in the Mtb genome), non-expressing genes (read counts in all samples were zero) as well as genes that had full genome deletions. As the isolates used for this analysis were the same isolates used for the benchmark, deletions in those isolates were known. Read counts were subsequently normalized using the trimmed mean of $M$ values (TMM) factor[88], and the TMM normalized reads per kilobase million (RPKM) were calculated using the edgeR package (v3.36.0)[89]. After TMM normalization, log2(RPKM+1) was calculated and defined as transcriptional expression levels. We used limma (v3.50.3)[90] for the subsequent steps: we created a multidimensional scaling plot of distances between gene expression profiles to visualize the transcription profiles (Supplementary Fig. 20). Expression values were fitted to a linear model for estimation of gene expression levels specific to each condition. To stabilize variance estimates and improve the detection of differentially expressed genes, empirical Bayes moderation was applied to the standard errors. Finally, significantly differentially expressed genes were identified by extracting genes with adjusted $p$ values <0.05 (Benjamin–Hochberg correction) with an absolute log2 fold change ≥1.

## SV genotyping of 44,709 Illumina-sequenced isolates

About 44,709 high-quality Illumina fastq files were downloaded from public repositories using fastq-dl. SVs were genotyped as described above using shovill, minigraph with the -l 10,000 and -d 5000 flags to account for more fragmented assemblies and miniwalk (in this case using the -na flag). We obtained assemblies and VCF files for 41,134 isolates. SVs were merged as done for the long-read data.

## Phylogeny of 41,134 isolates

SNPs for each sample were called using snippy (v4.6.0)[91]. A pairwise SNP matrix was done using psdm (v0.3.0)[92] and a phylogenetic tree was built using VeryFastTree (v4.0.4)[93].

## SV association analysis with DR

To account for population structure in the association analyses, SNPs for each of the 41,134 samples were called using snippy, non-SNP variants and SNPs falling in the masking sites detailed in Marin et al.[10] were filtered out of each VCF file. All VCF files were merged into one

using bcftools. SNPs with a minimum allele frequency <0.01 were filtered out, and a principal component analysis (PCA) was done using Hail (v0.2.13)[94]. The first four PCs accounted for 90.9% of the variance, so those were used in the association analyses. Each sample was also sent through TB-Profiler to account for those samples where DR was explained by SNVs or already-annotated SVs by the WHO[95]. Two models were done to find SVs associated with DR. First, a Firth logistic regression was done with the drug as the outcome and the SVs as the predictors with the 4 PCs and the TB-Profiler predictions as covariates. Firth was chosen over regular logistic regression to account for the class imbalance in our dataset (less resistant isolates than susceptible ones). Adding TB-Profiler DR predictions as a covariate ensured that associated SVs were independent from already-known WHOv2-listed or TB-Profiler resistance mutations (including already-listed SVs such as the *katG* deletion). *P* values were adjusted after each drug using the Benjamin–Hochberg correction. Associated SVs were filtered out if their adjusted *p* values were >0.05, their frequency in resistant isolates was less than double that of susceptible isolates, they were small deletions (≤100 bp) in complex regions[10], presented a minimum allele frequency (MAF) <0.01 in the resistant isolates or were phylogenetically restricted. The second approach was xgboost (v1.7.8.1)[96] as it was able to handle sparse data. As we were only interested in finding the features associated with each drug and not in creating a predictive model, we did not separate between testing and training datasets, allowing us to detect less frequent SVs associated with DR. As for the logistic regression model, the drug phenotype was used as the outcome and the SVs as predictors with the same covariates as the logistic regression model. Feature importance was then extracted from the resulting model. A third analysis was done for those SVs that were associated with various drugs, to determine which drug association was strongest, as there were various cases of MDR isolates. In those cases, the SV was used as the dependent variable while the drugs were the independent variables with the same covariates as before. To further confirm the role of the SVs as the sole mutations associated with the resistant phenotype, we assessed whether the isolates presenting those SVs, with no TB-Profiler resistance prediction, had SVs in known, resistance-conferring genes[5], to filter out SVs associated with DR-causing genes but not with DR (for example, large deletions spanning the resistance-causing gene and surrounding genes).

For the SV-gene burden approach, if an SV overlapped a gene, regardless of the position and length, those genes would be annotated as affected for those isolates with the SVs. The same pipeline as for the SV-based test was applied.

### Permutation analysis to assess statistical significance

To evaluate whether the associations observed between structural variants (SVs) and DR phenotypes could occur by chance, we performed a permutation analysis. For each variant/gene, we randomly shuffled the DR labels among the samples, breaking the true associations while preserving the dataset structure. We repeated this process 200 times, running the same association tests for each permutation to generate a null distribution of *p* values for each SV.

The observed *p* values from the actual dataset were then compared to this null distribution to calculate empirical *p* values, representing the proportion of permutations with *p* values as significant as or more significant than those observed. This analysis allowed us to assess the likelihood of finding these associations by chance.

### Statistics and visualizations

Unless stated otherwise, all statistics and visualizations were done in R.

### Reporting summary

Further information on research design is available in the Nature Portfolio Reporting Summary linked to this article.

## Data availability

ONT data generated in this study for African lineages (L5, L6, and L9) is available in BioProject PRJNA857537, while the other newly sequenced lineages can be found in Bioproject PRJNA1193213. All public long-read data is available in Supplementary Data 1. All RNA-Seq accession IDs are in Supplementary Tables 2 and 3. All public Illumina data used for this project can be found in https://doi.org/10.5281/zenodo.7819984. The Mtb-PRG and phylogenetic trees are available at https://doi.org/10.5281/zenodo.14842102. Source data can be found in https://doi.org/10.5281/zenodo.17216946.

## Code availability

Miniwalk can be accessed through https://github.com/aleixcanalda/miniwalk and the code used for this manuscript can be found in https://github.com/aleixcanalda/Mtb_PRG_Paper[97].

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

## Acknowledgements

We thank José Antonio Barraza López for assistance with figure design and image preparation.

## Author contributions

A.C.-B., M.S., M.B.H., L.C., and S.J.D. conceived this project. ONT sequencing was carried out by D.T., L.T.V. and N.L.S. All bioinformatics analyses and experiments were done by A.C.-B. M.B.H. computed the 44k isolates' phylogenetic tree. The work was supervised by M.S., M.B.H., X.C., L.C., and S.J.D. A.C.-B. created the first draft, which was modified and revised by all authors.

## Competing interests

The authors declare no competing interests.
