## [Transparent Peer Review file · Nature Communications]

Genome graphs reveal the importance of structural variation in *Mycobacterium tuberculosis* evolution and drug resistance

Corresponding Author: Professor Sarah Dunstan

Version 0:

Reviewer comments:

Reviewer #1

(Remarks to the Author)

Canalda-Baltrons et al developed miniwalk to genotype structural variants (SVs) from long and short reads. They evaluated the algorithm and found it achieves high genotyping accuracy. They then applied miniwalk to public *Mycobacterium tuberculosis* (Mtb) samples and investigated the effect of SVs in them. Overall, the algorithm is reasonable and the improvement over short-read SV calling is expected. I have some concerns over the data quality but I believe this would not affect the major conclusion of this manuscript.

1) The authors evaluated miniwalk on 16 genomes, taking SVIM-ASM as the ground truth. I am a little surprised by the low consistency in Fig. 1a, in particular the big drop in the range of 500-1000bp (Fig. 1b) where long-read SV discovery is supposed to have enough power. Have the authors compared the evaluation results to truvari? What are the main differences between SVIM-ASM and miniwalk? Is it caused by different SV representations, complex SVs, misalignment to wrong graph paths, or by low-frequency SVs not present in the graph? It would be good if the authors could show a few examples for readers to understand the cause.

2) QUASt often classifies real SVs to assembly errors. Strains diverged from the reference genome are more likely to fail the QUASt filtering. Could the authors show the number of misassemblies (Line 495) as a function of SNP/k-mer based distance from H37Rv? This would give readers an idea whether the threshold of 100 would cause biases.

3) The authors evaluated the genotyping algorithm, but they did not check the quality of SVs encoded in the graph. The authors removed 38 most fragmented assemblies (Line 140) but they had to remove additional 75 to generate a reasonable Fig. 2c. For consistency, I would suggest using the high-quality subset of 746 genomes (=859-38-75) in section 2.3 through 2.6, unless important strains are missing from this high-quality subset.

4) Along the line of quality control, I wonder if the authors could plot the allele frequency spectrum of SVs in Line 147 and show if singleton SVs (i.e. those occurring in one genome only) tend to come from fragmented assemblies. For common SVs, they may check the consistency with the phylogenetic tree of the 746 samples. Given ideal data, a true SV should be explained by a single mutation event; a false SV tends to occur in distant lineages. Incomplete assemblies will complicate this analysis. Restricting the analysis to high-quality assemblies may alleviate this concern. Fig. 2c is close to this analysis, but it is not locating the SVs to the internal branches if I am right.

5) Relatedly, the authors removed 10% of outliers in the ESX analysis (Line 210). Did they remove outlier genes or outlier SVs? In either case, are these outliers caused by false SVs? What would the numbers look like if they do not remove outliers? Would focusing on the high-quality subset of genomes reduce outliers?

6) In Fig. 6c, no SVs are perfectly linked to the predicted drug resistance (DR) status. There could be several explanations such as inaccuracy in TB-profiler or inaccuracy in SV genotyping. What are the individual genotyping accuracy of the SVs in Fig. 6c in the benchmark dataset? Could the authors manually check a few present/absent SV calls in this list?

7) Statistical significance discussed on Line 336 is not reflected in Fig. 6c. I would recommend showing four numbers: number of resistant samples with the SV, number of resistant samples without the SV, number of susceptible with the SV and number of susceptible without the SV (like the numbers we use for Fisher's exact test).

8) Line 151: could the authors check the boundaries of the translocation to see if IS6110 is involved?

(Remarks on code availability)

Reviewer #2

(Remarks to the Author)

(Remarks on code availability)

Reviewer #3

(Remarks to the Author)

Review – “Genome graphs reveal the importance of structural variation in *Mycobacterium tuberculosis* evolution and drug resistance”

The manuscript presents an impressive long-read catalogue of 859 *M. tuberculosis* genomes, a pangenome reference graph (Mtb-PRG), and a new graph-aware SV caller, miniwalk. Miniwalk achieves higher precision than a linear pipeline (manta), and the resulting SV atlas is mined for lineage-specific events (e.g. ESX-5, ctpV) and drug-resistance associations in 41,134 isolates. The study is methodologically innovative, biologically rich, and well documented. The points below—mostly clarifications or focused extensions—would make the work stronger.

Major comments:

1. The evaluation set used to benchmark miniwalk may be too narrow to yield an unbiased precision estimate. Only 16 polished long-read assemblies are excluded from the pangenome graph, whereas the remaining 843 assemblies are incorporated and share large stretches of sequence with the graph used for testing. In addition, lexicographic order may bias node representation toward early genomes. Both factors risk inflating performance. Re-ordering assemblies by phylogeny or allele frequency would test the robustness of graph structure.
2. The gold standard for evaluation is to use simulated rather than real data, where the performance can be tested against the truth.
3. The manuscript reports a 66-kb “inversion + translocation” in lineage 9, yet the Methods section states that miniwalk outputs only INS, DEL, DUP and INV. Please clarify exactly how this rearrangement was detected—did miniwalk generate it automatically, or was it reconstructed manually by inspecting walk paths? If miniwalk cannot output translocation, please indicate whether a post-processing script or future native support is planned.
4. Since the PRG excludes inserts ≤ 49 bp, miniwalk misses many micro-indels and under-performs in the 50–500 bp bin. Consider a data-specific threshold: keep the 49 bp cutoff for long-read assemblies (to avoid error noise) but drop it to e.g., 20 bp for short-read assemblies, then run a small test to report the recall gain versus any added false positives.
5. At present the link between the ctpV deletion and intracellular copper accumulation is currently supported only by transcriptomic differences from a single L1.2.1 isolate versus two controls. Do the authors plan adding functional evidence such as ICP-MS copper measurements, copper-stress growth assays, or genetic complementation to confirm that this deletion indeed alters Cu homeostasis and confers an adaptive benefit?
6. Although miniwalk is demonstrated on the single-chromosome *M. tuberculosis* genome, its utility for bacteria with multi-chromosome or large plasmids remains untested. Please discuss expected performance on two-chromosome bacteria or plasmid-rich species.

Minor comments:

I am not sure I have fully understood Fig. 3b.

In the table below the ESX-5 bubble, the haplotype that occurs in 9 % of isolates is labelled simply “PPE27”; the schematic shows that PPE27 is retained, whereas PPE25, PE18 and PPE26 are absent. Yet the main text (lines 198–199) refers to this event as “a ppe25-*ppe27* deletion,” which might be read as deleting all four genes. Could the authors clarify the exact boundaries of the common deletion and, if necessary, adjust the wording (e.g. “a deletion spanning the *ppe25*–*ppe27* locus that removes *ppe25*, *pe18* and *ppe26* while leaving *ppe27* intact”) so readers can unambiguously match the figure, the table and the text? A small schematic indicating retained vs. deleted segments in the legend would help.

(Remarks on code availability)

Version 1:

Reviewer comments:

Reviewer #1

(Remarks to the Author)

I thank the authors for the careful responses. I do not have further comments.

(Remarks on code availability)

Reviewer #3

(Remarks to the Author)

Review – “Genome graphs reveal the importance of structural variation in *Mycobacterium tuberculosis* evolution and drug resistance”

The authors addressed the main concerns (expanded hold-out to 32; lineage-ordered graph; simulation baseline; 20-bp cutoff test; L9 event clarified; Fig. 3b wording fixed; see Figs. S6, S7, S13, and S25). Results look robust.

Final suggestions:

1. Benchmark robustness — Report 95% confidence intervals for precision and recall on the 32-sample hold-out.
2. Simulation benchmark details — In the Supplement, state what dataset was simulated, which SV classes were inserted (e.g., INS/DEL/DUP/INV), and the exact counts per class (and total); a brief size-range summary would help.
3. Translocation evidence (revised): The Methods clearly describe the L9 case. For completeness, could you briefly note in Results/Discussion whether you observed any other INS-DEL pairs of similar size with high sequence identity between loci? If so, would you consider these candidate translocations (possibly with inversion), or interpret them as other mechanisms (like complex SVs)?
4. Applicability to multi-chromosome/plasmid-rich genomes — While miniwalk is expected to generalize via minigraph, this study did not evaluate multi-chromosome or plasmid-rich genomes. Please add a brief sentence in the Discussion noting this limitation and recommending current use on single-chromosome genomes, with multi-replicon evaluation left for future work.

With these minimal edits, I would support acceptance.

(Remarks on code availability)

Reviewer #1 (Remarks to the Author):

Canalda-Baltrons et al developed miniwalk to genotype structural variants (SVs) from long and short reads. They evaluated the algorithm and found it achieves high genotyping accuracy. They then applied miniwalk to public Mycobacterium tuberculosis (Mtb) samples and investigated the effect of SVs in them. Overall, the algorithm is reasonable and the improvement over short-read SV calling is expected. I have some concerns over the data quality but I believe this would not affect the major conclusion of this manuscript.

We thank the reviewer for their positive and constructive feedback which we believe has helped strengthen our manuscript.

Below, we address each of the reviewer's specific concerns and suggestions in detail.

1) The authors evaluated miniwalk on 16 genomes, taking SVIM-ASM as the ground truth. I am a little surprised by the low consistency in Fig. 1a, in particular the big drop in the range of 500-1000bp (Fig. 1b) where long-read SV discovery is supposed to have enough power. Have the authors compared the evaluation results to truvari? What are the main differences between SVIM-ASM and miniwalk? Is it caused by different SV representations, complex SVs, misalignment to wrong graph paths, or by low-frequency SVs not present in the graph? It would be good if the authors could show a few examples for readers to understand the cause.

We thank the reviewer for this comment. The observed drop in consistency for the 500–1000 bp range in Fig. 1b indeed prompted further investigation.

Upon closer examination, we found that this drop was primarily driven by a single complex SV that was consistently detected by miniwalk but missed by SVIM-ASM. Specifically, miniwalk reported a total of 40 SVs in this size range, 20 of which were labeled as false positives (FPs), resulting in a precision of 0.5. Notably, 11 of these 20 FPs originated from the same SV, which was present in 11 of the 16 assemblies. This variant alone substantially contributed to the reduced precision in this bin.

Miniwalk consistently identified this event as a 620 bp deletion preceded by a 2,083 bp insertion, whereas SVIM-ASM did not report any structural variation across the region despite evidence from both the assembly and long-read alignments supporting the presence of a variant (Fig. S5). This suggests that miniwalk may have greater sensitivity for complex SVs.

If we consider this 620 bp deletion as a true positive, the precision for this size range increases substantially from 0.5 to 0.78.

Moreover, we did not assess this inconsistency using other tools, such as truvari, as manually inspecting the long-read and assembly alignments helped resolve the issue.

As clarified above, the use of a linear genome could hinder an SV caller's ability to call SVs (<https://doi.org/10.1038/s41586-023-05896-x>), which is what differentiates SVIM-ASM from miniwalk. This has been further explained in the Methods section (Line 582):

“SVIM-ASM was chosen as it is a well-established assembly SV caller reliant on a linear genome, which is what differentiates it from minigraph/miniwalk”

We have now updated the Results section (Line 93) to clarify this point and provide a more detailed explanation for the reduced precision observed in the 500–1,000 bp range:

“Nevertheless, precision varied across SV types, with DELs of size 500-1,000bp having the lowest precision (0.5; Fig.1b), though fewer SVs were found in that size range. Further analysis revealed that this drop was largely driven by a single complex SV present in 11 assemblies, consistently detected by miniwalk but missed by SVIM-ASM (Fig.S5)”

Fig.S5 IGV visualisation of the complex SV considered as a false positive compared against the SVIM-ASM SV callset. The top track represents the assembly alignment while the bottom

track represents the long-read alignment to H37Rv. These tracks show that the SV is not a false positive and is present in both types of alignments, despite SVIM-ASM not detecting it.

2) QUILT often classifies real SVs to assembly errors. Strains diverged from the reference genome are more likely to fail the QUILT filtering. Could the authors show the number of misassemblies (Line 495) as a function of SNP/k-mer based distance from H37Rv? This would give readers an idea whether the threshold of 100 would cause biases.

We thank the reviewer for this helpful suggestion. The QUILT misassembly filter was applied primarily to exclude a limited number of lower-quality PacBio (non-HiFi) assemblies. In total, this affected 24 assemblies, most of which were generated using older PacBio (PB) technologies and consistently underperformed across multiple quality metrics. As a result, we opted to exclude all non-HiFi PacBio assemblies from the analysis entirely.

Given this, the misassembly filter served only as a preliminary safeguard screening and had no practical impact on the set of assemblies included in downstream analyses. Removing the filter would not alter our results, as these assemblies were already excluded based on sequencing platform and overall quality.

Nevertheless, we still thought it would be important to examine this concern from the reviewer about potential bias from QUILT's misassemblies metric, so we calculated the k-mer genetic distance of each assembly to H37Rv using Mash (excluding the PB assemblies). We observed a negative correlation (p-value = $7e-9$), indicating that more divergent assemblies were not more likely to be flagged with higher misassembly counts. This suggests that the QUILT filter did not introduce systematic bias against genetically distant strains.

Given that the non-HiFi PacBio assemblies were excluded independently of this filter, and that no positive correlation was observed between divergence and misassembly count, we conclude that the 100-misassembly threshold did not meaningfully affect the final dataset or bias the results. This analysis has now been added to the Methods section of the manuscript (Line 521):

“Correctness was assessed by applying a number of misassemblies filter of maximum 100. While this threshold excluded only 24 assemblies, all of them generated with PacBio CLR (non-HiFi) data, we chose to exclude all PacBio CLR assemblies from the analysis due to their consistently poor quality across multiple metrics. To evaluate potential bias introduced by the misassembly filter, we examined the relationship between k-mer-based distance to H37Rv (via Mash (v2.3)) and QUILT misassembly counts. We observed a negative correlation (p-value = $7e-9$; Fig.S24), indicating that more divergent strains were not disproportionately affected by the misassembly threshold.”

Fig.S24 The number of misassemblies reported by QUASt as a function of k-mer-based genetic distance from the reference genome H37Rv.

3) The authors evaluated the genotyping algorithm, but they did not check the quality of SVs encoded in the graph. The authors removed 38 most fragmented assemblies (Line 140) but they had to remove additional 75 to generate a reasonable Fig. 2c. For consistency, I would suggest using the high-quality subset of 746 genomes (=859-38-75) in section 2.3 through 2.6, unless important strains are missing from this high-quality subset.

We thank the reviewer for this thoughtful suggestion. We agree that consistency in filtering is important and aimed to maintain it where appropriate. However, the additional 75 assemblies excluded in Fig. 2c were removed specifically to improve the clarity of the SNP-based phylogenetic tree, due to known issues with SNP calling in older long-read-only assemblies (<https://www.nature.com/articles/s41598-025-90089-x>). These assemblies were not removed due to low quality or unreliable SV content. Since the structural variant genotyping pipeline is independent of the SNP-based tree reconstruction, we retained the dataset of 821 genomes for the analyses in Sections 2.3-2.6 to ensure maximum diversity and statistical power. We believe that the additional quality control provided in comments 4-5 suggests that the SVs from the 821 assemblies are reliable, including those from the 75, excluded assemblies from the phylogeny. Furthermore, we verified that the assemblies

excluded from the tree do not show higher SV counts or lower assembly metrics, supporting their inclusion in SV genotyping analyses.

Moreover, the majority of isolates that were excluded from the tree were in the correct major lineage (L1-9), except a group of isolates falling in L4.9, meaning most excluded isolates had erroneous sub-lineage positioning, which indicates these isolates still had enough high-quality SNPs to be correctly placed in their respective main lineage.

Therefore our evidence suggests that removing the 75 assemblies would not significantly change the results of sections 2.3-2.6.

Here we are showing the tree without removing the 75 isolates. Around 20 isolates are incorrectly placed at the lineage level, while the other 50 isolates are incorrectly placed at the sub-lineage level (incorrect sub-lineage assignment cannot be visualised here).

We have clarified this issue in the Methods section (Line 636):

“To improve the accuracy and interpretability of the SNP-based phylogeny, we excluded 75 assemblies that produced discordant lineage calls and phylogeny placement. This was likely the result of using earlier long-read technologies, which are associated with higher SNP calling error rates (Carandang et al., 2025)”

4) Along the line of quality control, I wonder if the authors could plot the allele frequency spectrum of SVs in Line 147 and show if singleton SVs (i.e. those occurring in one genome only) tend to come from fragmented assemblies. For common SVs, they may check the consistency with the phylogenetic tree of the 746 samples. Given ideal data, a true SV should be explained by a single mutation event; a false SV tends to occur in distant lineages. Incomplete assemblies will complicate this analysis. Restricting the analysis to high-quality assemblies may alleviate this concern. Fig. 2c is close to this analysis, but it is not locating the SVs to the internal branches if I am right.

We thank the reviewer for this important suggestion regarding the potential spurious nature of singleton SVs. In total, we identified 3,077 structural variants, of which 1,617 (52.6%) were singletons, i.e. observed in only one assembly (Fig.S10a).

To assess whether these singleton SVs are associated with fragmented or lower-quality assemblies, we examined the number of singletons in the 75 phylogeny-excluded assemblies, assemblies with an N50 below 2Mb and in assemblies with an N50 > 2Mb. We found that singleton SVs are not disproportionately associated with fragmented or the phylogeny-excluded assemblies, and therefore are unlikely to be systematic artifacts resulting from poor assembly quality (Fig.S10b). The high percentage of singletons also indicates that there is likely more structural variation to discover across the *Mtb* phylogeny. Moreover, this further confirms the high SV quality of the 75 phylogeny-excluded assemblies. This has been added to the Results (Line 170):

“The allele frequency spectrum of SVs revealed that over half (1,617/3,077) were singletons (Fig.S10a). Importantly, singleton SVs were not enriched in fragmented assemblies, suggesting they represent genuine rare variation rather than artifacts (Fig.S10b).”

Fig.S10 **a** Allele frequency spectrum of the 3,077 SVs. **b** Violin plot of the number of singleton SVs found in each assembly. Assemblies are separated into those excluded from the phylogeny in Fig.1c, those not excluded from the phylogeny with an N50 < 2Mb and those with an N50 > 2Mb.

To assess the quality of common SVs, we found that 24% of them were found exclusively inside a single lineage. We also found that the most frequent SVs (VAF > 0.85) were present in all lineages and clades except L4.9 (H37Rv's lineage), as would be expected. We randomly checked SVs with variant allele frequencies ranging 0.01-0.5 and found they

consistently fell in clades that made phylogenetic sense, with certain exceptions for SVs in complex regions or transposable elements.

Nevertheless, SVs don't always behave like SNPs in the phylogeny, as SVs in repetitive regions (i.e. the ESX-5 deletion) have much higher mutation rates. A previous study found that the rate of indels in repetitive regions is at least an order of magnitude greater than that of SNVs (<https://doi.org/10.1073/pnas.2301394120>). For this reason, examining homoplasy isn't a reliable way of running QC on SV calls.

5) Relatedly, the authors removed 10% of outliers in the ESX analysis (Line 210). Did they remove outlier genes or outlier SVs? In either case, are these outliers caused by false SVs? What would the numbers look like if they do not remove outliers? Would focusing on the high-quality subset of genomes reduce outliers?

We thank the reviewer for pointing out the possibility of our ESX analysis being confounded by false SVs. We removed outlier genes that had an extremely high amount of SVs across all isolates. This also excluded a single SV event that was present in almost all isolates, except L4.9, as that was the reference genome. With the current analysis, the mean number of SVs per non-essential gene (excluding outliers) is 0.76 and 0 in essential genes. Without removing outliers, this number goes up to 17 for non-essential genes and 4 for essential genes. The SVs that fall in most essential genes do not alter the gene sequence as they are duplications or deletions in a repetitive region, with the resulting sequence remaining intact (the case for *rplW*, an essential gene with SVs in 328 isolates, all located at the 3' end of the gene, with the resulting SVs maintaining the STOP codon intact). That is why we believe that these genes are exceptions and are not representative of most essential genes that do not have any SVs in them. As for non-essential genes, large PE/PPE genes can have more than 1 SV in a single isolate, which brings the number of SVs across our whole dataset up to 2,232 instances of SVs across all isolates in *PPE34*. Such a gene also represents an exception and extremely biases the mean. Moreover, the PE/PPE genes are known to fall in complex regions of the genome, prone to presenting structural variations.

Re-doing the analysis without the 75 isolates, excluded from the phylogeny, does not change the results. Without removing outliers, we had an average number of SVs per gene of 3.8 for essential genes and 16.47 for non-essential genes, the slight reduction likely due to having a smaller dataset rather than removing spurious SVs.

We have clarified this issue in the Results (Line 227):

"... we calculated the trimmed (10%; removal of outlier **genes with high SV counts**) mean number of SVs in essential (0) and non-essential (0.76) genes..."

6) In Fig. 6c, no SVs are perfectly linked to the predicted drug resistance (DR) status. There could be several explanations such as inaccuracy in TB-profiler or inaccuracy in SV genotyping. What are the individual genotyping accuracy of the SVs in Fig. 6c in the benchmark dataset? Could the authors manually check a few present/absent SV calls in this list?

Out of the 14 SVs associated with drug resistance, only 2 are found in the 16 assemblies used for the benchmark (1414787 90bp DEL and 2208005 12,720bp DEL). The 12,720bp DEL had a precision of 100% (was found in only the short-read assemblies where it really existed), whereas the 90bp deletion was not found in the short-read assemblies benchmark (i.e. that SV was a false negative).

We decided to manually check for the presence/absence of other SVs in a few isolates:

- 830274 DEL 3109bp

Here we are showing in the top track short-read data mapped to the H37Rv reference genome from an isolate with the 3,109bp SV (SV called by miniwalk) and susceptibility to isoniazid. The bottom track shows an isolate, also susceptible to isoniazid without an SV (no SV called by miniwalk).

- 2288842 DEL 353bp

The top track represents the *pncA* deleted region (SV called by miniwalk) in a pyrazinamide-resistant (PZA-DR) isolate, the middle track represents an isolate with no *pncA* deletion (no SV called by miniwalk) nor PZA-DR and the bottom track represents a *pncA* deleted region (SV called by miniwalk) in a pyrazinamide-susceptible isolate. The differences in SV sizes are due to the low frequency of *pncA* SV representations in the *Mtb*-PRG, highlighting the need for more long-read DR isolates to capture more SV diversity. Nevertheless, the *Mtb*-PRG correctly predicted the presence of a deletion in both susceptible and resistant isolates.

- 2531901 DEL 265bp

The top track represents an isolate without capreomycin resistance (CAP-DR) nor the predicted deletion (no SV called by miniwalk). The middle track represents an CAP-DR isolate with the miniwalk-predicted deletion. The bottom track represents a capreomycin-susceptible isolate with the miniwalk-predicted deletion.

This deletion is found within a complex, repetitive region; the accurate detection with miniwalk in both resistant and susceptible isolates further confirms the robustness of our pipeline.

In all isolates that were manually checked, the SVs were concordant with their miniwalk predictions. Moreover, we used the TB-Profiler predictions not as our dependent variable but as a covariate, to account for SNVs and SNPs that were contributing to the drug resistance phenotypes. Therefore, the lack of resistant-unique SVs was not due to the TB-Profiler predictions.

We believe the more likely explanation for the absence of SVs uniquely present in drug-resistant isolates is that our binary classification of drug resistance may be too coarse to capture the underlying biology. Many of the SVs we identified may not independently confer full resistance, but instead contribute to an increased minimum inhibitory concentration (MIC), which would not be reflected in binary resistance calls. Future studies that use MIC as a continuous phenotype, rather than a binary label, may be better positioned to detect SVs associated with incremental changes in drug susceptibility.

Our findings here are supported by a previous study (<https://doi.org/10.1099/mgen.0.001081>) that examined deletions in canonical drug resistance genes, revealing that only deletions affecting *katG* were unique to resistant isolates. Deletions in *pncA* (associated with pyrazinamide resistance) and *ethA* (ethionamide resistance), however, were observed in both resistant and susceptible isolates, consistent with what we observe in our dataset. Moreover, SNPs strongly associated with drug resistance in the WHO catalogue have also been found in susceptible isolates ([https://doi.org/10.1016/s2666-5247\(21\)00301-3](https://doi.org/10.1016/s2666-5247(21)00301-3)).

We discuss this possibility in our Discussion (Line 464):

“We hypothesize that the lack of DR-unique SVs in our results may reflect the limitations of binary resistance labels, as many SVs may influence minimum inhibitory concentration (MIC) without conferring full resistance. As shown in recent studies, using MIC instead of binary phenotypes might enhance low-level DR variants detection.”

7) Statistical significance discussed on Line 336 is not reflected in Fig. 6c. I would recommend showing four numbers: number of resistant samples with the SV, number of resistant samples without the SV, number of susceptible with the SV and number of susceptible without the SV (like the numbers we use for Fisher's exact test).

We appreciate the reviewer’s suggestion and have now modified Fig.6c:

Drug	Start	Length	Type	Resistant samples with the SV	Resistant samples without the SV	Susceptible samples with the SV	Susceptible samples without the SV	Gene/s
INH	3853488	59	DEL	1189	12173	11	23836	Rv3434c
PZA	1695684	3265	DEL	176	3081	48	13399	Rv1507c, Rv1505c, Rv1506c, Rv1507A, Rv1508c
ETO	4326900	1655	DEL	131	2535	26	10116	ethR, Rv3856c, ethA
PZA	2288842	353	DEL	38	3524	3	16862	pncA
STR	3878595	220	INS	33	2546	30	4782	
INH	830274	3109	DEL	735	10422	20	19627	Rv0738, Rv0739, Rv0741, PE_PGRS8, Rv0740
PZA	2287446	2968	DEL	38	3524	5	16860	Rv2041c, pncA, lipT, Rv2042c, Rv2044c
PZA	1414787	90	DEL	81	2996	55	13820	pknH
CFZ	364642	3972	DEL	2	143	6	9475	Rv0303, PPE5, Rv0302
CAP	2531901	265	DEL	77	682	437	8440	
DLM	964686	261	DEL	18	70	550	6262	rpfA
DLM	4367556	2111	DEL	7	55	172	6144	mycP2, eccE2
DLM	2531901	265	DEL	9	66	226	6097	
CFZ	2208005	12720	DEL	2	138	28	8859	Rv1974, Rv1975, Rv1976c, Rv1972, yrbE3A, Rv1973, mce3B, mce3D, lprM, mce3F, yrbE3B, mce3A, mce3C, Rv1977

8) Line 151: could the authors check the boundaries of the translocation to see if IS6110 is involved?

While IS6110 being involved in the large L9 translocation was an interesting hypothesis, we did not find evidence of that being the case. The translocated region was originally around 2,900,000-2,967,000, and, after doing a BLAST of the IS6110 sequence against the H37Rv genome, we only found an IS6110 element at 2,970,000 but none close to 2,900,000. The closest was around 2,780,000, likely too far away to have been involved in the translocation.

Reviewer #2 (Remarks to the Author):

Reviewer #3 (Remarks to the Author):

Review – “Genome graphs reveal the importance of structural variation in *Mycobacterium tuberculosis* evolution and drug resistance”

The manuscript presents an impressive long-read catalogue of 859 *M. tuberculosis* genomes, a pangenome reference graph (Mtb-PRG), and a new graph-aware SV caller, miniwalk. Miniwalk achieves higher precision than a linear pipeline (manta), and the resulting SV atlas is mined for lineage-specific events (e.g. ESX-5, ctpV) and drug-resistance associations in 41,134 isolates. The study is methodologically innovative, biologically rich, and well documented. The points below—mostly clarifications or focused extensions—would make the work stronger.

We thank the reviewer for their thoughtful and encouraging feedback. We are also grateful for the reviewer’s constructive suggestions for clarification and extension, which we believe have strengthened the manuscript.

Below, we provide detailed responses to each of the reviewer’s specific points.

Major comments:

1. The evaluation set used to benchmark miniwalk may be too narrow to yield an unbiased precision estimate. Only 16 polished long-read assemblies are excluded from the pangenome graph, whereas the remaining 843 assemblies are incorporated and share large stretches of sequence with the graph used for testing. In addition, lexicographic order may bias node representation toward early genomes. Both factors risk inflating performance. Re-ordering assemblies by phylogeny or allele frequency would test the robustness of graph structure.

We thank the reviewer for this comment regarding the potential for bias in our benchmarking strategy due to overlap between the pangenome graph and test assemblies. We agree that such overlap could lead to optimistic performance estimates if not properly accounted for.

To evaluate this concern, we conducted an additional analysis in which we excluded 32 assemblies—double the original number—from the pangenome graph. These additional exclusions were chosen to exclude at least an isolate from major lineages (1–7). We then re-ran our benchmarking analysis on this expanded hold-out set.

Encouragingly, the performance remained consistent: mean precision was 0.84 (vs 0.85, two-sided t-test $p = 0.55$) and mean recall was 0.80 (vs 0.82, two-sided t-test $p = 0.15$), showing no statistically significant difference from our original evaluation. This suggests that the initial benchmark, despite potential graph overlap, provided a robust estimate of real-world performance.

More broadly, we agree with the reviewer that benchmarking on completely novel genomes is an important test. However, we would also emphasize that the practical goal of miniwalk is to genotype structural variation in new samples that are expected to share large segments of DNA with previously characterized isolates. In this context, the presence of common variation in the graph is not a confounder, but rather a design feature that enhances genotyping accuracy. Our results support the method's effectiveness under both realistic and more stringent exclusion scenarios.

We also appreciate the reviewer's point about potential bias from assembly ordering during graph construction. To address this concern, we re-ordered the graph input assemblies by lineages, instead of lexicographic ordering (excluding the 16 held-out assemblies), and recalculated the precision, recall and F1-score. A t-test comparing the precision of the original *Mtb*-PRG against the sorted PRG showed no statistically significant difference (two-sided t-test p -value=0.96, sorted PRG precision mean=0.85). We also compared the recall values and found no statistically significant difference (double-sided t-test p -value=0.29, sorted PRG recall mean=0.81).

Fig.S6 Results comparing the phylogenetically-sorted *Mtb*-PRG and miniwalk SV genotyping against SVIM-ASM truth SV calls using the 16 long-read, polished assemblies. These results show no significant difference from Fig.1a.

These results show that the initial graph structure is robust to genome input variation, with no significant variation in genotyping accuracy. This analysis has been added to the Results (Line 99):

“To test whether assembly ordering introduced bias, we rebuilt the *Mtb*-PRG with assemblies ordered by lineage rather than lexicographic filename and re-evaluated SV genotyping performance. Precision and recall did not differ significantly (two-sided t-tests; precision $p=0.96$, recall $p=0.29$), indicating that graph construction order had no measurable effect on genotyping accuracy (Fig.S6)”

2. The gold standard for evaluation is to use simulated rather than real data, where the performance can be tested against the truth.

We appreciate the reviewer’s suggestion and agree that simulation-based benchmarking provides the advantage of a known ground truth, making it an attractive approach for controlled performance evaluation. In response, we explored this avenue by generating a simulated dataset in which we inserted SVs derived from one real assembly into a reference

genome and then simulated a long-read assembly using vcflib (<https://github.com/vcflib/vcflib>).

This simulation allowed us to evaluate performance under idealized conditions, yielding a precision of 0.92 and recall of 0.82—substantially higher than the 0.78 precision and 0.81 recall observed when benchmarking on the original real assembly. These results initially seemed encouraging. However, upon deeper analysis and in line with prior findings (<https://doi.org/10.1186/s12864-022-08548-y>; <https://doi.org/10.1186/s12864-016-2366-2>), we observed that simulated SVs do not fully capture the structural complexity, reference ambiguity, and assembly artifacts characteristic of real pathogen genomes.

Consequently, while we found simulation a valuable complementary tool for establishing baseline performance, it tends to overestimate real-world accuracy. For this reason, we ultimately prioritized evaluation on high-quality long-read-based SVs from real assemblies, which more accurately reflect the challenges inherent in real-world pathogen surveillance and epidemiology. Nonetheless, the simulation results support the robustness of our method under ideal conditions and serve as a useful supplement to our primary benchmarks.

We have added the simulated benchmark to the Results (Line 103) and Methods (Line 601).

Results:

“To complement real-data benchmarking, we simulated an assembly by spiking real SVs into the H37Rv reference using vcflib. Miniwalk achieved 0.92 precision and 0.82 recall, outperforming results on real data (0.78/0.81), highlighting the reduced complexity of simulations.”

Methods:

“For simulation benchmarking, we used vcflib to introduce SVs from a real assembly into the H37Rv reference, then simulated a long-read assembly for evaluation. This setup provides a controlled test with known ground truth while maintaining realistic SV profiles.”

3. The manuscript reports a 66-kb “inversion + translocation” in lineage 9, yet the Methods section states that miniwalk outputs only INS, DEL, DUP and INV. Please clarify exactly how this rearrangement was detected—did miniwalk generate it automatically, or was it reconstructed manually by inspecting walk paths? If miniwalk cannot output translocation, please indicate whether a post-processing script or future native support is planned.

We thank the reviewer for pointing this out. The 66-kb "inversion + translocation" event in lineage 9 was not output directly by miniwalk, but was instead manually identified via visual

inspection in IGV. We dove deeper into this SV as the translocation represented the largest INS and DEL in our dataset, both unique to L9. As noted, miniwalk currently supports INS, DEL, DUP, and INV, consistent with its reliance on minigraph, which does not natively identify or represent translocations (<https://github.com/lh3/minigraph/issues/68>). Therefore, minigraph’s inability to handle translocations limits their detection through miniwalk.

Detecting translocations automatically would require sequence content comparison across distant loci, which is particularly challenging in *M. tuberculosis* due to the abundance of transposable elements and high homology between mobile regions. Because of this complexity, we did not implement automated translocation detection, nor do we currently plan to in the scope of this work.

However, we believe this finding is still biologically important and have revised the manuscript to clarify that this rearrangement was manually reconstructed and confirmed, rather than directly output by miniwalk. We have updated the Methods section accordingly (Line 612):

“L9 translocation detection

The largest SV identified in our dataset was a ~66,000 bp rearrangement in a L9 isolate, involving both an inversion and a translocation into a novel genomic position. Since minigraph does not natively represent translocations in the graph structure, such events are fragmented across non-adjacent nodes. As a result, miniwalk reported this event as two separate SVs of similar size—an INS and a DEL—without resolving the rearrangement as a translocation (Fig.S25).

To investigate the true nature of this event, we aligned the L9 long reads to the H37Rv reference using minimap2. Manual inspection in IGV revealed that reads mapping to the insertion breakpoint also aligned to the distant deletion site, consistent with a translocation. Furthermore, reads spanning the region corresponding to the deleted sequence were mapped in the reverse orientation, indicating that the translocated segment was also inverted in its new genomic location.”

Fig.S25a

Fig.S25b IGV visualisation of the deleted (a) and inserted (b) sites. Long reads spanning the inserted and deleted breakpoints are also shown.

4. Since the PRG excludes inserts ≤ 49 bp, miniwalk misses many micro-indels and underperforms in the 50–500 bp bin. Consider a data-specific threshold: keep the 49 bp cutoff for long-read assemblies (to avoid error noise) but drop it to e.g., 20 bp for short-read assemblies, then run a small test to report the recall gain versus any added false positives.

We thank the reviewer for this insightful suggestion. As recommended, we lowered the insertion length threshold from 50 bp to 20 bp, created a new *Mtb*-PRG and benchmarked

the 16 short-read assemblies against the SVIM-ASM long-read SVs. Overall, the precision decreased approximately 2% (from 0.7 to 0.68) while recall increased 2% (from 0.34 to 0.36). Specifically, for SVs with sizes 50-500bp, the precision decreased 0.5% while recall increased 3%. More notably, recall increased 4% for INSS, at the expense of an 8% decrease in precision. However, this was driven by small INSS. INSS > 1,000bp had a decreased precision of 2% with an increase in recall of 4%.

We have added this observation into the Results (Line 124):

“To explore whether recall could be improved, we created another *Mtb*-PRG, retaining SVs ≥ 20 bp, and evaluated it against the short-read assemblies. We observed, on average, a 2% increase in recall at the cost of an 2% drop in precision (Fig.S7).”

Fig.S7 Histogram with 50 bp bins showing the found SVs genotyping the short-read assemblies using miniwalk and the *Mtb*-PRG with SVs ≥ 20 bp, against the SVIM-ASM SVs. We show two segments with 50 bp bin from SVs of size 50-100 bp, 400 bp bin for SVs of size 100-500 bp, 500 bp bin for SVs of size 500-3,000 bp and SVs larger than 3,000 bp grouped together. The two segments show the precision and recall across different SV sizes.

5. At present the link between the *ctpV* deletion and intracellular copper accumulation is currently supported only by transcriptomic differences from a single L1.2.1 isolate versus two controls. Do the authors plan adding functional evidence such as ICP-MS copper

measurements, copper-stress growth assays, or genetic complementation to confirm that this deletion indeed alters Cu homeostasis and confers an adaptive benefit?

We thank the reviewer for highlighting this important point. We agree that functional assays, such as ICP-MS copper measurements, copper-stress growth experiments or genetic complementation, would provide valuable confirmation of the role of the *ctpV* deletion in copper homeostasis. However, performing such experiments is outside the scope of the current study.

Nonetheless, the observed *ctpV* deletion, coupled with transcriptomic alterations in copper-responsive genes, provides a strong hypothesis for future functional investigation. We have now added a statement to the Discussion (Line 441) to explicitly acknowledge this limitation and propose targeted experimental validation as a priority for follow-up work:

“A limitation of the supporting RNA-Seq data is that it was generated in only 1 isolate for each sub-lineage, which may not represent the transcriptional profile of all isolates in those sub-lineages. To overcome this and directly test the functional consequences of the *ctpV* deletion, future studies should include intracellular copper quantification, copper-stress growth assays, and genetic complementation experiments. Additionally, animal infection models that support granulomatous lesions, where copper levels are elevated under hypoxic conditions, could provide further insight into whether this deletion confers an adaptive advantage *in vivo*”

6. Although miniwalk is demonstrated on the single-chromosome *M. tuberculosis* genome, its utility for bacteria with multi-chromosome or large plasmids remains untested. Please discuss expected performance on two-chromosome bacteria or plasmid-rich species.

We thank the reviewer for this insightful comment. While miniwalk has been demonstrated on the single-chromosome genome of *M. tuberculosis*, we agree that its applicability to bacteria with multiple chromosomes or large plasmid content is an important consideration. As miniwalk complements the output of minigraph, which supports multiple chromosomes, we expect that miniwalk will remain effective in these contexts as well. However, we acknowledge that empirical validation on such genomes remains to be done and represents a valuable direction for future work. This has now been addressed in the Discussion (Line 398):

“As miniwalk complements minigraph's output, it could be used with graphs from other species, allowing for wider applications. In particular, while miniwalk's performance on multi-chromosome bacteria or plasmid-rich species remains untested, we expect it to be

similarly effective in these contexts, as minigraph is capable of handling multiple chromosomes”

Minor comments:

I am not sure I have fully understood Fig. 3b.

In the table below the ESX-5 bubble, the haplotype that occurs in 9 % of isolates is labelled simply “PPE27”; the schematic shows that PPE27 is retained, whereas PPE25, PE18 and PPE26 are absent. Yet the main text (lines 198–199) refers to this event as “a ppe25–ppe27 deletion,” which might be read as deleting all four genes. Could the authors clarify the exact boundaries of the common deletion and, if necessary, adjust the wording (e.g. “a deletion spanning the ppe25–ppe27 locus that removes ppe25, pe18 and ppe26 while leaving ppe27 intact”) so readers can unambiguously match the figure, the table and the text? A small schematic indicating retained vs. deleted segments in the legend would help.

We thank the reviewer for pointing out the confusion our wording may generate. Indeed, the deletion spans all four genes, however, as the start of both ppe25 and ppe27 are homologous, the resulting ppe27 gene remains intact. This deletion has been previously described (<https://doi.org/10.1128/jb.00827-08>). This can also be seen in Fig.S13. We have modified the explanation of this deletion in the Results (Line 218):

“The deletion can most likely be attributed to non-allelic homologous recombination as the 3' sequences of *ppe25* and *ppe27* are highly homologous (Fig.S13), that is, the deletion spans the *ppe25–ppe27* locus and removes *ppe25*, *pe18* and *ppe26* while leaving *ppe27* intact”

We have also modified Fig.3b to show the deletion that is happening more clearly:

Reviewer #1 (Remarks to the Author):

I thank the authors for the careful responses. I do not have further comments.

Reviewer #3 (Remarks to the Author):

Review – “Genome graphs reveal the importance of structural variation in *Mycobacterium tuberculosis* evolution and drug resistance”

The authors addressed the main concerns (expanded hold-out to 32; lineage-ordered graph; simulation baseline; 20-bp cutoff test; L9 event clarified; Fig. 3b wording fixed; see Figs. S6, S7, S13, and S25). Results look robust.

We are glad the reviewer is satisfied with our work thus far.

Below, we provide detailed responses to each of the reviewer’s specific points.

Final suggestions:

1. Benchmark robustness — Report 95% confidence intervals for precision and recall on the 32-sample hold-out.

The 95% confidence intervals for the 32-sample hold-out were 0.814-0.860 for the precision and 0.781-0.822 for the recall.

2. Simulation benchmark details — In the Supplement, state what dataset was simulated, which SV classes were inserted (e.g., INS/DEL/DUP/INV), and the exact counts per class (and total); a brief size-range summary would help.

We have now added this information into the Supplement:

“

3. Simulation dataset

The dataset we used for the simulation benchmark was derived from the N1216 *Mtb* isolate's VCF obtained by mapping to the reference genome H37Rv and calling SVs using

SVIM-ASM. We therefore added 2 INVs (1,621-2,181bp), 34 DELs (51-17,582bp) and 35 INSS (53-5,000bp).

”

We refer to this additional information in the Methods section of our manuscript:

“For simulation benchmarking, we used vclib to introduce SVs from a real assembly into the H37Rv reference, then simulated a long-read assembly for evaluation (more details in Supplementary Methods 3).”

3. Translocation evidence (revised): The Methods clearly describe the L9 case. For completeness, could you briefly note in Results/Discussion whether you observed any other INS-DEL pairs of similar size with high sequence identity between loci? If so, would you consider these candidate translocations (possibly with inversion), or interpret them as other mechanisms (like complex SVs)?

We thank the reviewer for this suggestion.

In addition to the L9 translocation, we identified more INS-DEL pairs with high sequence identity within each isolate. These may represent candidate translocations, especially since they have the same size and share high sequence similarity to the IS6110 transposable element, though we cannot exclude alternative mechanisms such as complex SVs.

We have added this observation to our Results:

“The translocation implicated 70 genes and truncated *Rv2577* and *Rv0345*. This rearrangement highlights the complexity of SV patterns observed in *Mtb*, though most isolates presented small DELs, CNVs and numerous IS6110 elements (1,358bp; Fig.2a). We found various INS-DEL pairs of similar size ($\geq 90\%$) and high identity ($\geq 90\%$) across each genome (range of 0-3 per genome) which could be translocations, likely stemming from IS6110 transposition.”

4. Applicability to multi-chromosome/plasmid-rich genomes — While minigraph is expected to generalize via minigraph, this study did not evaluate multi-chromosome or plasmid-rich genomes. Please add a brief sentence in the Discussion noting this limitation and recommending current use on single-chromosome genomes, with multi-replicon evaluation left for future work.

We agree with the reviewer that, as our tool has only been tested in a single-chromosome context, we should address this limitation and highlight multi-replicon evaluation for future work:

“As miniwalk complements minigraph's output, it could be used with graphs from other species, allowing for wider applications. In particular, while miniwalk's performance on multi-chromosome bacteria or plasmid-rich species remains untested, we expect it to be similarly effective in these contexts, as minigraph is capable of handling multiple chromosomes. However, given that this study did not evaluate multi-chromosome or plasmid-rich genomes, we recommend applying miniwalk primarily to single-chromosome genomes at present, with extension to multi-replicon contexts left for future work.”

With these minimal edits, I would support acceptance.